# Dual midbrain and forebrain origins of thalamic inhibitory interneurons

**Polona Jager[1], Gerald Moore[2,3], Padraic Calpin[4], Xhuljana Durmishi[1], Irene Salgarella[1], Lucy Menage[1], Yoshiaki Kita[5], Yan Wang[5], Dong Won Kim[6], Seth Blackshaw[6], Simon R Schultz[2], Stephen Brickley[3], Tomomi Shimogori[5], Alessio Delogu[1]\***

[1]Department of Basic and Clinical Neuroscience, Institute of Psychiatry, Psychology and Neuroscience, King's College London, London, United Kingdom; [2]Department of Bioengineering, Imperial College London, London, United Kingdom; [3]Department of Life Sciences and Centre for Neurotechnology, Imperial College London, London, United Kingdom; [4]Department of Physics and Astronomy, University College London, London, United Kingdom; [5]RIKEN, Center for Brain Science (CBS), Saitama, Japan; [6]The Solomon H. Snyder Department of Neuroscience, School of Medicine, Johns Hopkins University, Baltimore, United States

**Abstract** The ubiquitous presence of inhibitory interneurons in the thalamus of primates contrasts with the sparsity of interneurons reported in mice. Here, we identify a larger than expected complexity and distribution of interneurons across the mouse thalamus, where all thalamic interneurons can be traced back to two developmental programmes: one specified in the midbrain and the other in the forebrain. Interneurons migrate to functionally distinct thalamocortical nuclei depending on their origin: the abundant, midbrain-derived class populates the first and higher order sensory thalamus while the rarer, forebrain-generated class is restricted to some higher order associative regions. We also observe that markers for the midbrain-born class are abundantly expressed throughout the thalamus of the New World monkey marmoset. These data therefore reveal that, despite the broad variability in interneuron density across mammalian species, the blueprint of the ontogenetic organisation of thalamic interneurons of larger-brained mammals exists and can be studied in mice.

**\*For correspondence:**
alessio.delogu@kcl.ac.uk

**Competing interests:** The authors declare that no competing interests exist.

## Introduction

The thalamus is a forebrain structure that develops from the diencephalic prosomere 2 (p2) (*Puelles and Rubenstein, 2003*; *Shi et al., 2017*; *Wong et al., 2018*) and is primarily composed of cortically projecting excitatory thalamocortical (TC) neurons, divided into more than 30 individual nuclei in mammals (*Clascá et al., 2012*; *Hunnicutt et al., 2014*; *Jones, 2007*). The function of the thalamus has been historically described as relay of sensory information to the cortex (*Cheong et al., 2013*; *Hubel and Wiesel, 1962*; *Piscopo et al., 2013*; *Shatz, 1996*; *Sherman and Guillery, 2002*; *Van der Loos and Woolsey, 1973*; *Zeater et al., 2015*). Taking into account the diversity of input and output features of TC circuits (*Clascá et al., 2012*; *Guillery, 1995*; *Herkenham, 1980*; *Rubio-Garrido et al., 2009*; *Sherman, 2016*), more recent work has shown that the thalamus is also critically involved in cognitive processes allowing for behavioural flexibility (*Bolkan et al., 2017*; *Groh et al., 2014*; *Guo et al., 2017*; *Ling et al., 2015*; *Rikhye et al., 2018a*; *Rikhye et al., 2018b*; *Saalmann and Kastner, 2011*; *Schmitt et al., 2017*; *Sherman, 2016*).

In contrast to cortical networks, excitatory neurons in the thalamus do not connect with each other (*Bickford et al., 2008*; *Hirsch et al., 2015*; *Jones, 2007*; *Rikhye et al., 2018b*). Instead, local connections and computations within TC circuits are dominated by the resident inhibitory, GABA-

releasing neurons (interneurons) (*Hirsch et al., 2015*; *Montero, 1987*; *Pasik and Pasik, 1976*; *Sherman, 2004*).

Interneuron numbers and distribution vary widely across species, suggesting that they are critically involved in the evolution of TC structure and function (*Arcelli et al., 1997*; *Letinic and Rakic, 2001*; *Rikhye et al., 2018b*). In particular, comparative studies across all amniotes (reptiles, birds, and mammals) have described a correlation between the proportion of interneurons and the size and connectivity of the excitatory thalamus (*Arcelli et al., 1997*; *Butler, 2008*).

For example, in the reptilian thalamus, which is mostly devoid of descending projections from the cortex, interneurons have only been identified in the retinorecipient regions (*Butler, 2008*; *Kenigfest et al., 1995*; *Kenigfest et al., 1998*; *Pritz and Stritzel, 1994*; *Rio et al., 1992*). In birds, however, where reciprocal connections between the thalamus and the cortex are more abundant, thalamic interneurons are distributed more widely (*Butler, 2008*; *Granda and Crossland, 1989*; *Veenman and Reiner, 1994*).

Similarly among mammals, interneurons are largely restricted to the visual thalamus in smaller-brained marsupials, bats and mice, where they represent only 6% of the total neuronal population (*Butler, 2008*; *Evangelio et al., 2018*; *Seabrook et al., 2013b*). In primates, on the other hand, where higher order (HO) nuclei driven by cortical inputs are expanded relative to sensory relay (first order, FO) regions (*Armstrong, 1979*; *Baldwin et al., 2017*; *Butler, 2008*; *Halley and Krubitzer, 2019*; *Stephan et al., 1981*), interneurons are present across the entire thalamus and their proportion increases to around 30% (*Arcelli et al., 1997*; *Braak and Bachmann, 1985*).

To what extent these differences are the result of species-specific ontogenesis of thalamic interneurons remains poorly understood. We have previously shown that in the mouse, interneurons in the FO visual thalamus, the dorsal lateral geniculate nucleus (LGd), originate in the midbrain from an $En1^+Gata2^+Otx2^+Sox14^+$ lineage (*Jager et al., 2016*). On the other hand, earlier work in humans has suggested the DLX1/2-expressing ganglionic eminences (GE) in the telencephalon as the source of interneurons for some of the larger HO thalamic nuclei – the mediodorsal (MD) nucleus and the pulvinar (*Letinić and Kostović, 1997*; *Letinic and Rakic, 2001*; *Rakić and Sidman, 1969*). At the same time, the latter studies were not able to detect any such migration from the GE in the mouse and macaque brain (*Letinic and Rakic, 2001*). While these findings therefore point to innovation in developmental origins, we currently lack an understanding of the shared ontogeny of mammalian thalamic interneurons.

Here we hypothesised that a blueprint of the complex organisation of thalamic interneurons observed in large-brained mammals is present even in the simpler thalamus of the mouse. This prediction is supported by findings from the cortex demonstrating that its inhibitory interneuron classes, generated in the subpallium and defined through expression of regulatory programmes (i.e. transcription factors), are common to the amniote lineages (*Arendt et al., 2019*; *Metin et al., 2007*; *Tasic et al., 2018*; *Tosches et al., 2018*). Moreover, a conserved subpallial origin was demonstrated for cortical interneurons in the cyclostome hagfish, and therefore appears to be an ancestral feature of the vertebrate brain (*Sugahara et al., 2017*; *Sugahara et al., 2016*).

Using genetic fate mapping, two-photon whole brain tomography, and spatial modelling, we investigated the ontogeny and distribution of thalamic GABAergic interneurons comprehensively across the mouse TC nuclei. We then used fluorescent in situ marker detection to compare the distribution of genetically defined interneuron classes in the New World marmoset monkey brain. These experiments identify in the mouse a wider distribution of GABAergic interneurons than previously reported (*Arcelli et al., 1997*; *Evangelio et al., 2018*; *Seabrook et al., 2013b*), encompassing both FO sensory relay and HO thalamic nuclei, including associative HO nuclei that are most enlarged in primates. We then show that while the largest proportion of thalamic interneurons in the mouse is generated in the $En1^+Sox14^+$ embryonic midbrain, there is an additional class that derives from the $Nkx2.1^-Lhx6^-Dlx5^+Foxd1^+$ inhibitory progenitor domains in the forebrain. Intriguingly, we also find that in the mouse interneurons are organised in a spatial pattern according to their ontogeny, such that midbrain-born interneurons are largely found in the sensory relays, while the forebrain-generated interneurons reside in the HO thalamus, including MD, laterodorsal (LD), and lateral posterior (LP; aka pulvinar) nuclei. Genoarchitectonic evidence supports a conserved basic organisation of thalamic interneurons in the non-human primate marmoset thalamus, where putative midbrain-generated interneurons are abundant in FO and HO nuclei and complemented by a distinct and more restricted interneuron class enriched in selected HO associative TC nuclei.

# Results

## *Sox14*-expressing interneurons are widely distributed across the FO and HO mouse thalamus

In the mouse thalamus, GABAergic interneurons are most abundant in the LGd (*Arcelli et al., 1997*; *Evangelio et al., 2018*). We had previously demonstrated that all LGd interneurons are defined by expression of the transcription factor gene *Sox14* and presented the *Sox14$^{GFP/+}$* knockin mouse line (*Table 1*; *Crone et al., 2008*) as a useful tool to study these cells (*Jager et al., 2016*). Both the Allen Brain Atlas (2015 Allen Institute for Brain Science. Allen Cell Types Database. Available from: cell-types.brain-map.org) and DropViz resources (available from: dropviz.org; *Saunders et al., 2018*) identify a *Sox14$^+$* transcriptional cluster corresponding to mouse LGd interneurons, confirming our previous findings. *Sox14* is expressed upon cell-cycle exit within inhibitory lineages in the diencephalon, midbrain, hindbrain, and spinal cord, but not in the telencephalon (*Achim et al., 2013*; *Delogu et al., 2012*; *Guo and Li, 2019*; *Prekop et al., 2018*).

To investigate the spatial distribution of *Sox14* neurons comprehensively across all thalamic (TC) regions in the mouse, we took advantage of the endogenous and bright fluorescence of GFP in postnatal day (P) 21 *Sox14$^{GFP/+}$* mice to perform high resolution (0.54 µm voxel) whole brain imaging by two-photon laser scanning tomography; for an indicative low-resolution scan through a series of z-projections, see also: *Video 1*. Optical sections were acquired in the coronal plane 10 µm apart and registered with the Allen Institute Common Coordinate Framework (CCF3; *Figure 1A*) using a custom Python pipeline alongside the Elastix registration library (*Klein et al., 2010*) to delineate anatomical subdivisions according to the Allen Brain Institute taxonomy (*Figure 1A* and *Video 2*). To perform automated detection of cells, a deep learning strategy was implemented to train a U-Net segmentation model (*Ronneberger and Brox, 2015*) on a data set of 12,264 images (512 × 512 pixels at 0.54 µm voxel), obtained by supplemental augmentation of 219 manually annotated samples (*Figure 1B*). The accuracy of our automated counting strategy was validated by comparing the total count of GFP$^+$ cells in the LGd of P21 *Sox14$^{GFP/+}$* mice (*Figure 1C,D*; 1234 ± 82; mean ± SD) to a recent stereological study of LGd GABA interneurons in the wild-type adult C57Bl/6 mouse (1255 ± 195; mean ± SD) (*Evangelio et al., 2018*), which confirmed the validity of our protocol.

**Table 1.** Animal models used in the study.

| Species | Designation | Source or reference | Identifiers | Additional information |
|---|---|---|---|---|
| *Mus musculus* | *Sox14$^{tm1Tmj}$* (*Sox14$^{GFP}$*) | *Crone et al., 2008* | MGI ID: 3836003 | Maintained in the C57BL/6J (Charles River Laboratories) background |
| *Mus musculus* | *En1$^{Cre}$* | *Kimmel et al., 2000*; The Jackson Laboratory | Stock No: 007916 MGI ID: 2446434 | C57BL/6J background |
| *Mus musculus* | *Dlx5/6$^{Cre}$* | *Monory et al., 2006*; The Jackson Laboratory | Stock No: 008199; MGI ID:3758328 | C57BL/6J background |
| *Mus musculus* | *Pvalb$^{Cre}$* | *Hippenmeyer et al., 2005*; The Jackson Laboratory | Stock No: 017320; MGI ID:3590684 | C57BL/6J background |
| *Mus musculus* | *Nkx2.1$^{Cre}$* | *Xu et al., 2008* | Stock No: 008661 MGI: J:131144 | C57BL/6J background |
| *Mus musculus* | *Foxd1$^{EGFPcre}$* (*Foxd1$^{Cre}$*) | *Humphreys et al., 2008* | Stock No: 012463 MGI:4359653 | C57BL/6J background |
| *Mus musculus* | *Lhx6$^{Cre}$* | *Fogarty et al., 2007* | Stock No: 026555 MGI:4355717 | C57BL/6J background |
| *Mus musculus* | RCE:loxP (*Rosa26$^{lsl-GFP}$*) | *Sousa et al., 2009*; The Jackson Laboratory | MMRRC Stock No: 32037-JAX MGI:4412373 | C57BL/6J background |
| *Mus musculus* | Gt(ROSA)26Sortm5(CAG-Sun1/sfGFP)Nat (*Rosa26$^{lsl-nuclearGFP}$*) | *Mo et al., 2015*; The Jackson Laboratory | Stock No: 021039; MGI ID: 5443817 | C57BL/6J background |
| *Callithrix jacchus* | WT | Colony at RIKEN RRD | - | - |

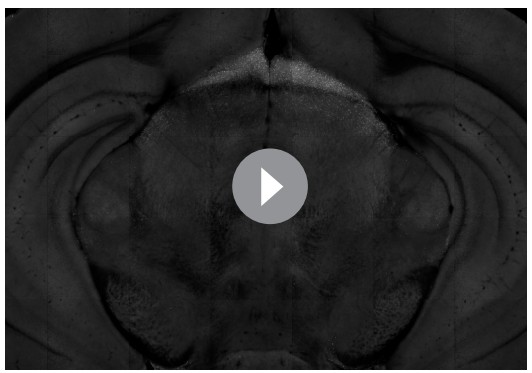

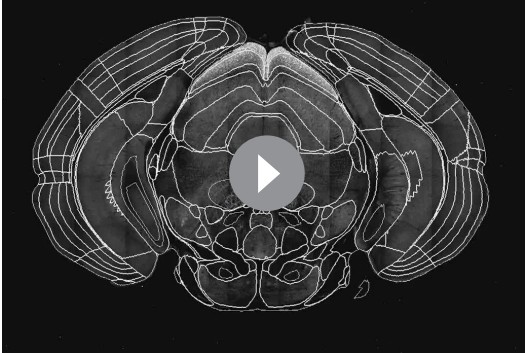

**Video 1.** A video animation of 28 z-stacks (each 100 µm) of projected coronal slices, downsized to 1 µm voxel size in XY to reduce file size. Each slice is a maximum intensity projection across 10 serial two-photon optical slices.

https://elifesciences.org/articles/59272#video1

**Video 2.** A video animation of the anatomical delineations from the Allen Institute Common Coordinate Framework (CCF3) projected onto the imaging data from a P21 $Sox14^{GFP/+}$ brain. Isotropic 10 µm voxel size.

https://elifesciences.org/articles/59272#video2

Automated counting identified a total of 6588 ± 811 (mean ± SEM) GFP$^+$ cells across TC nuclei of both hemispheres spanning the rostrocaudal and mediolateral extensions of the thalamus (n = 3 $Sox14^{GFP/+}$ at P21). Their distribution was not stochastic but skewed instead towards sensory TC nuclei and within sensory modality by a prevalence in FO nuclei. The GFP$^+$ cells are most abundant in the visual FO LGd (1234 ± 82) and HO LP (aka pulvinar; 411 ± 110), followed by sensory-motor ventrobasal (VB) complex [ventral posteromedial (VPM; 250 ± 49), ventral posterolateral (VPL; 105 ± 40), ventral anterior lateral (VAL; 4 ± 3) and ventral medial nucleus (VM; 4 ± 2)] and HO posterior nucleus (PO; 99 ± 12), in turn followed by the auditory FO ventral medial geniculate (MGv; 53 ± 12) and HO dorsal medial geniculate nucleus (MGd; 20 ± 13). Sparse GFP$^+$ cells were also detected in limbic TC nuclei: LD (20 ± 7), parafascicular (PF) (25 ± 5), and MD (13 ± 6) nuclei. Notably, GFP$^+$ neurons are typically found in the caudal-most part of the LD, nearer the LP border and nearly all GFP$^+$ neurons were in the caudal half of the MD (see also: *Video 1*). Counts per TC nucleus are reported for a single hemisphere.

### *Sox14* expression distinguishes between two spatially clustered interneuron classes

To validate the inhibitory nature of the GFP$^+$ cells in TC nuclei of the $Sox14^{GFP/+}$ mouse, we combined immunodetection of GFP$^+$ with in situ RNA hybridisation (ISH) against the *Gad1* mRNA in $Sox14^{GFP/+}$ mice sampling the rostrocaudal extent of the thalamus at approximately 200 µm intervals (*Figure 2A–C*). In our experience, the simultaneous detection of protein and mRNA is more reliable in younger tissue, hence experiments were done at P14, by which time point mouse TC nuclei are considered to display adult-like circuit composition (*Bickford et al., 2010*; *Golding et al., 2014*; *Seabrook et al., 2013a*; *Seabrook et al., 2013b*; *Thompson et al., 2016*).

Recapitulating the distribution observed at P21, GFP$^+$ cells were detected in the LGd, LP, VP, and PO and in very small numbers in the MG (*Figure 2A,C,D*). In these nuclei all GFP$^+$ cells had a GABAergic profile and co-expressed *Gad1* (100%, n = 3 brains). In the LGd, VP, and MGv (i.e. FO sensory relay nuclei) they also represented virtually all GABAergic cells (≥98%, pie charts in *Figure 2D*).

Unexpectedly, however, 22.1 ± 4.0% of the total GABAergic population in TC regions did not express GFP (*Figure 2B,C*; Figure 5Bii), and these GFP$^-$*Gad1*$^+$ cells appeared spatially largely non-overlapping with the *Sox14*$^+$ interneuron class and enriched at more rostral TC territory (*Figure 2C, D*). *Gad1*$^+$ cells are seen populating these rostral TC regions during the first five postnatal days (*Figure 2—figure supplement 1*). In particular, we observed that the distribution of GFP$^-$*Gad1*$^+$ cells is skewed towards the limbic HO MD (29.6 ± 4.5%) and LD (19.4 ± 3.1%) and the associative HO LP (13.2 ± 1.2%), and in smaller numbers towards the PO. Sparse GFP$^-$*Gad1*$^+$ cells were also found in thalamic regions where nuclear boundaries cannot be defined precisely at this age and that contain

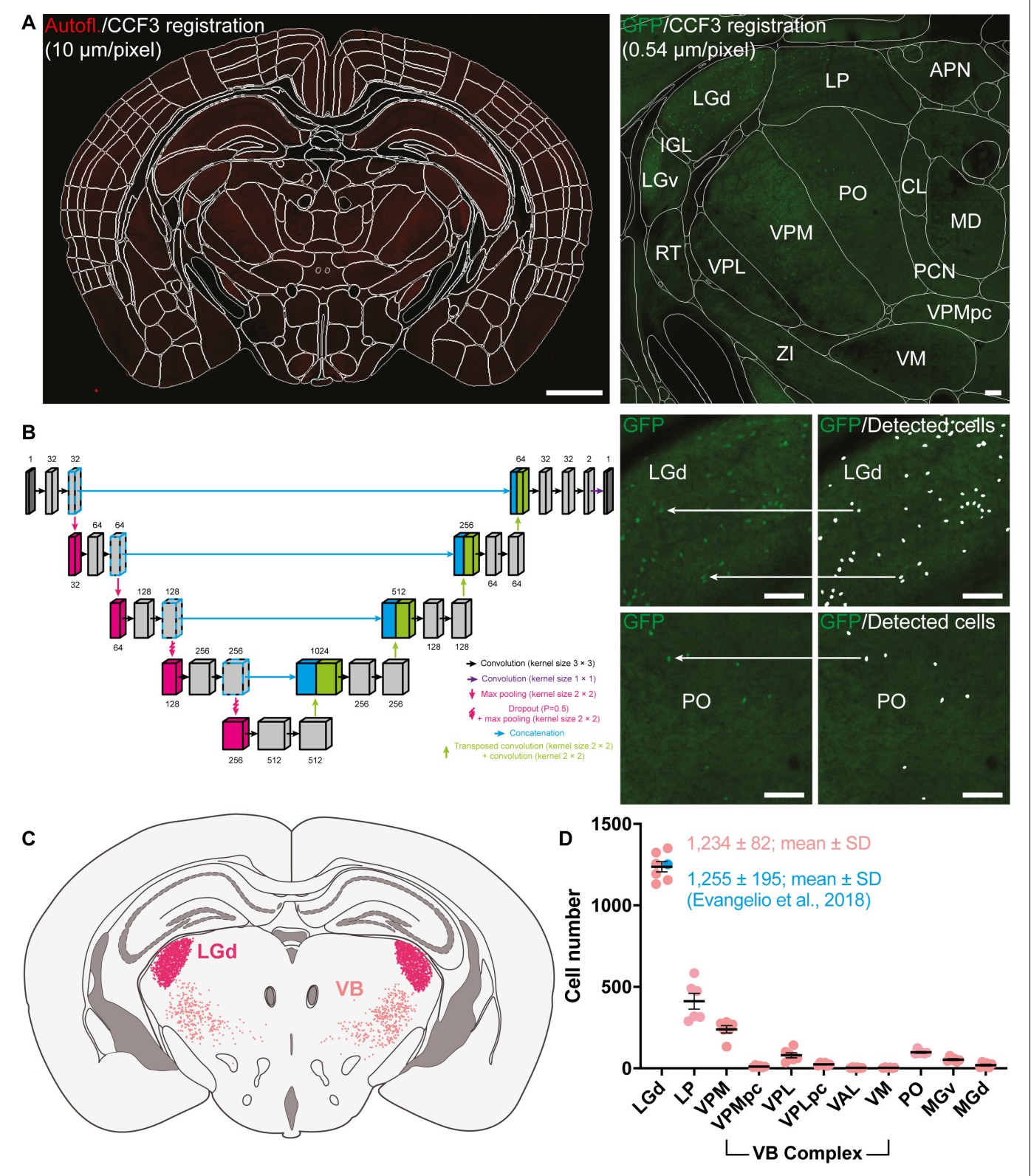

**Figure 1.** Automated total counts of GFP+ cells in the thalamus of the *Sox14GFP/+* mouse. (**A**) Autofluorescence (Autofl.) from serial two-photon imaging of Sox14GFP/+ mice (n = 3) at 0.54 × 0.54 × 10 μm voxel resolution was registered to the Allen Institute CCF3 atlas using Elastix (left; scale bar 1 mm). This permits delineation and identification of all anatomical structures according to the Allen Institute hierarchical taxonomy (right; scale bar 100 μm). (**B**) Automated cell detection was done using a U-Net trained on 219 manually segmented images (512 × 512 pixels) augmented to a total sample size

*Figure 1 continued on next page*

*Figure 1 continued*

of 12,264, split 75% for training and 25% validation. Images containing GFP fluorescence were passed into the trained U-Net (left) for cell prediction based on learned features during training (right; scale bar 100 μm). Oversampling in the z-axis was corrected for by grouping and averaging detected cell positions which colocalised within a defined cell radius. (C) Example illustration of automatically detected cells in the LGd and VB complex projected onto a representative coronal section of the thalamus. (D) Quantification of GFP$^+$ cells in the LGd at 1234 ± 82 (mean ± SD) validated against stereological study by *Evangelio et al., 2018* of 1255 ± 195 (mean ± SD) interneurons in the LGd. Other counts are shown for LP, VB complex [VPM, parvicellular part of the ventral posteromedial nucleus (VPMpc), VPL, parvicellular part of the ventral posterolateral nucleus (VPLpc), VAL, VM], MGv, MGd, and PO.

VAL, VM, centrolateral (CL), intermediodorsal (IMD), PF, reuniens (RE), romboid (RH), subparafascicular (SPF), subparafascicular area (SPA), central medial (CM), and anteromedial (AM) nuclei. Cell counts in these regions were grouped together under 'other TC' (*Figure 2D*).

To quantitatively demonstrate spatial clustering of these two putative thalamic interneuron classes (*Gad1$^+$Sox14$^+$* and *Gad1$^+$Sox14$^-$*), we calculated the nearest neighbour distances (NND) from 3D reconstructions of their respective distributions in the *Sox14$^{GFP/+}$* thalamus (*Figure 2C* and *Figure 3A*). Indeed, the cumulative distribution of NNDs was significantly shifted to smaller distances within each of the classes than between them (p<1.4 × 10$^{-30}$, two-sample Kolmogorov–Smirnov test, n = 3 brains; *Figure 3A*).

To characterise spatial organisation of thalamic GABAergic interneurons in an unbiased way, we then applied machine learning (k-means clustering) to these same 3D reconstructions of the *Sox14$^{GFP/+}$* thalami (*Figures 2C* and *3B,C*). The data best fit two spatial clusters, as assessed from the silhouette score (*Figure 3Bii,C*; see also Materials and methods). Consistent with the NND analysis, one cluster corresponded to the *Sox14$^+$* cells (contains 94.9 ± 1.4% of all *Sox14$^+$* cells), and the other to the *Sox14$^-$* interneurons (contains 81.0 ± 0.3% of all *Sox14$^-$* cells; *Figure 3B,D*). The two thalamic molecular GABAergic groups therefore occupy their own respective spatial clusters, with the *Sox14$^-$* cells located more rostrally and medially compared to the *Sox14$^+$* interneurons.

To independently confirm our findings and control for potential effects of looking at a juvenile age (P14), we also mapped anatomical distribution of all *Gad1$^+$* and *Chrna6$^+$* cells across the adult mouse TC nuclei at P56, using the Allen Mouse Brain Atlas (2004 Allen Institute for Brain Science. Allen Mouse Brain Atlas. Available from: mouse.brain-map.org; *Lein et al., 2007*) ISH data (*Figure 3—figure supplement 1A*). *Chrna6* has been identified as another marker specific for interneurons, at least in the LGd (*Golding et al., 2014*; DropViz; Allen Cell Types Database). The resulting 3D reconstructions, k-means spatial clustering (*Figure 3—figure supplement 1A*), and distribution plot (*Figure 3—figure supplement 1B*) were consistent with our observations from the P14 *Sox14$^{GFP/+}$* thalamus.

The mouse thalamus therefore exhibits wider interneuron diversity than has been previously reported, with at least two molecularly and spatially distinct classes. The largest interneuron class is defined by *Sox14$^+$* and is enriched in the caudal part of the thalamus which contains principal sensory relays and their associated HO nuclei. Conversely, the smaller *Sox14$^-$* GABAergic population is enriched in the rostral part of the thalamus, in HO regions that associate with more cognitive functions, such as the MD and LD (*Halassa and Kastner, 2017*; *Rikhye et al., 2018b*).

## The *Sox14$^+$* interneuron class is abundant and widespread in the marmoset thalamus

Given the sparseness of interneurons in the mouse thalamus, there exists the possibility that the *Sox14$^+$* interneuron class may represent a unique feature of smaller-brained species, or that it may be a conserved, but numerically negligible type of interneuron complemented by novel and more abundant types in species with larger brains. To detect the presence and assess the relative abundance of the *Sox14$^+$* interneuron class in the thalamus of species with a high density of interneurons, we sampled the distribution of *SOX14$^+$GAD1$^+$* cells in selected TC nuclei of the neonatal non-human primate marmoset.

Fluorescent ISH for *SOX14* and *GAD1* mRNAs revealed the widespread presence of *SOX14$^+$-GAD1$^+$* bona fide interneurons across all major TC nuclei (*Figure 4A–C*). Reminiscent of the expression pattern observed in the mouse, *SOX14$^+$* cells were not present in prethalamic structures [RT,

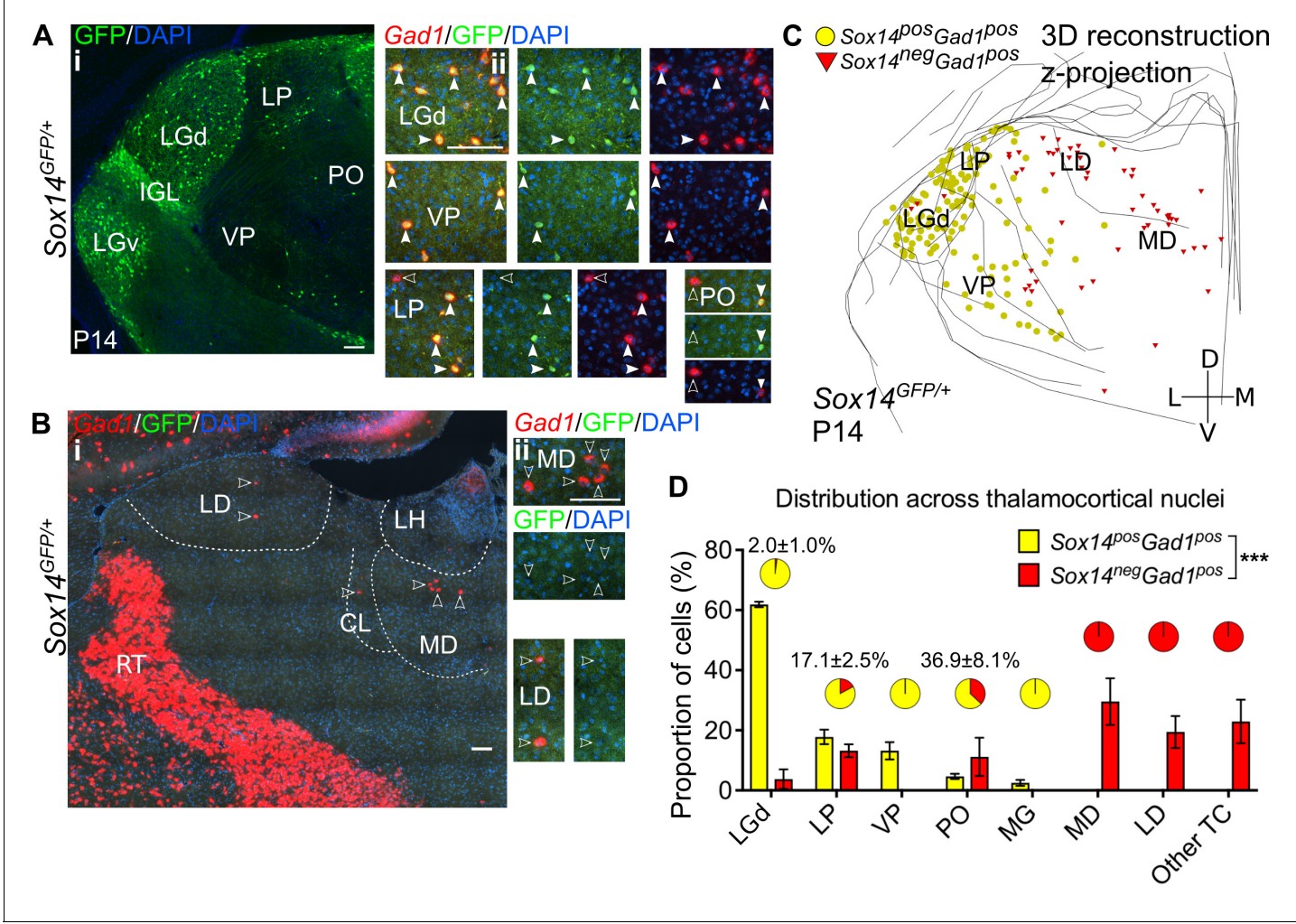

**Figure 2.** Diversity and distribution of GABAergic cells in the mouse thalamocortical nuclei. (A) (i) Representative coronal section of P14 *Sox14*<sup>GFP/+</sup> thalamus with *Sox14*⁺ cells in the LGd, VP, LP, and PO. (ii) *Sox14*⁺ cells in TC regions co-express *Gad1*, but not all *Gad1*⁺ cells co-express *Sox14* in the LP and PO. Filled arrows mark *Sox14*⁺*Gad1*⁺ and empty arrows *Sox14*⁻*Gad1*⁺ cells. Scale bars, 100 μm. (B) (i) Representative rostral coronal section of P14 *Sox14*<sup>GFP/+</sup> thalamus with *Gad1*⁺ cells in the MD, CL, and LD, which contain no *Sox14*⁺ cells. (ii) *Gad1*⁺ cells in these nuclei do not co-express *Sox14*. Scale bars, 100 μm. (C) 3D reconstruction of a representative P14 *Sox14*<sup>GFP/+</sup> thalamus from tracing every tenth 20-μm-thick coronal section, displayed as a z-projection and showing distribution of *Sox14*⁺*Gad1*⁺ (yellow) and *Sox14*⁻*Gad1*⁺ cells (red). One dot represents one neuron. (D) Distribution of *Sox14*⁺*Gad1*⁺ and *Sox14*⁻*Gad1*⁺ cells across TC nuclei in the *Sox14*<sup>GFP/+</sup> brains at P14, plotted as proportion of all the cells within each interneuron group (mean ± SEM; n = 3 brains). The category 'other TC' refers to regions where nuclear boundaries cannot be defined precisely and that contain VAL, VM, CL, IMD, PF, RE, RH, SPF, SPA, CM, and AM. *Sox14*⁺*Gad1*⁺ and *Sox14*⁻*Gad1*⁺ populations have distinct distributions (p<10⁻³ chi-squared test). Pie charts show the proportion (mean ± SEM) of the two interneuron classes within each nucleus.

The online version of this article includes the following figure supplement(s) for figure 2:

**Figure supplement 1.** Perinatal distribution of *Gad1*⁺ cells in the anterior thalamus.

zona incerta (ZI)], but detectable in the pregeniculate/subgeniculate (SubG), the primate homologue of the mouse IGL (*Figure 4B*).

Qualitative analysis shows largely overlapping distribution of the fluorescent probes for *SOX14* and *GAD1* in all TC nuclei at caudal (*Figure 4Ai–vi*) and intermediate levels (*Figure 4Bi–iii*), but some areas of differential expression at rostral level, where *GAD1* expression is not accompanied by *SOX14* expression in medial and dorsal regions of the thalamus (*Figure 4C*) that contain the limbic HO TC nuclei LD (*Figure 4Ci*) and MD (*Figure 4Cii*). We then proceeded to quantify the number of *SOX14*⁺*GAD1*⁺ and *SOX14*⁻*GAD1*⁺ cells in three brains of the new-born marmoset by randomly selecting three regions of 263 μm by 263 μm within the following nuclei: FO visual LGd, HO visual

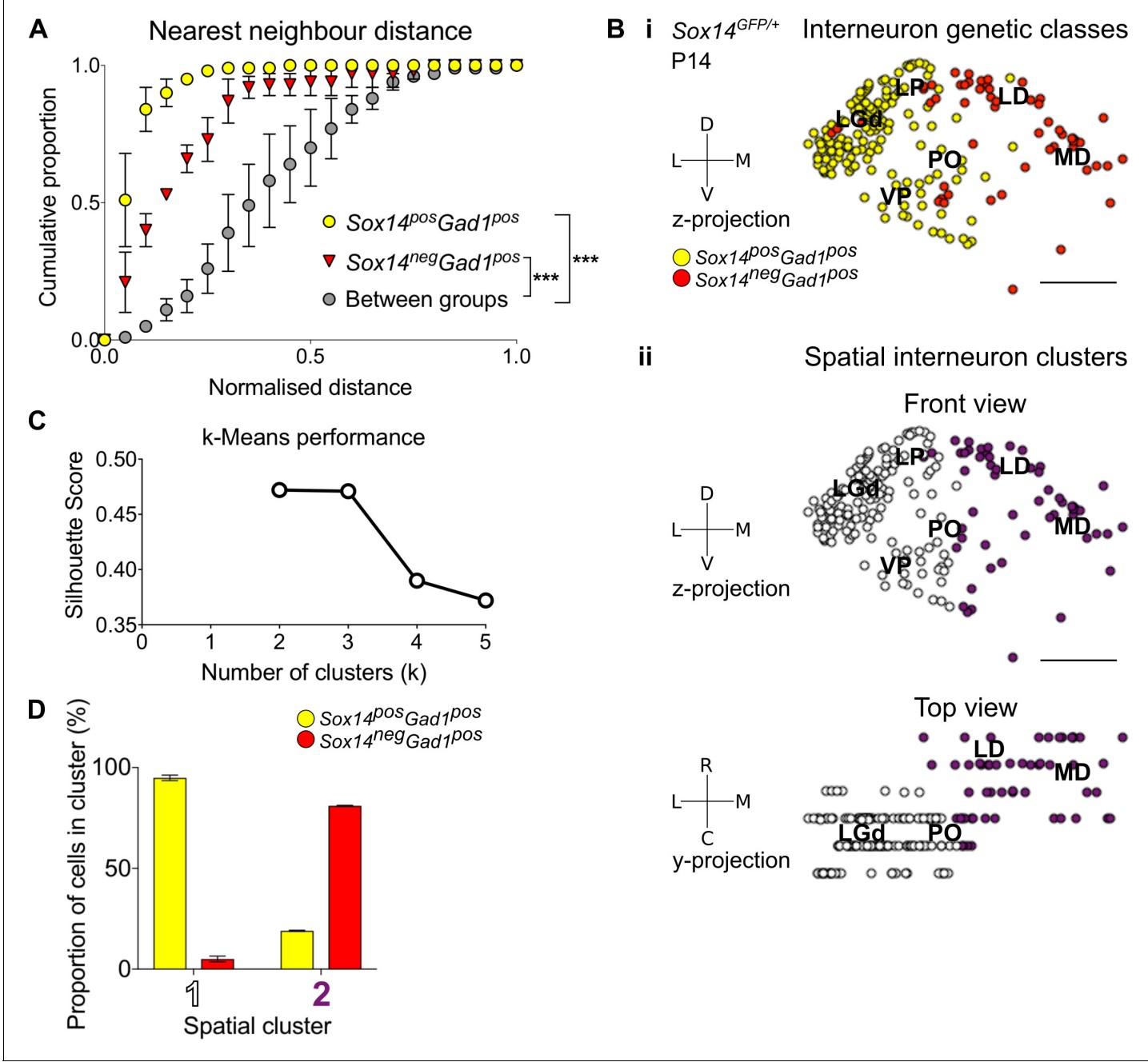

**Figure 3.** Spatial organisation of thalamic GABAergic cells. (**A**) Normalised nearest neighbour distance (NND) for *Sox14⁺Gad1⁺* and *Sox14⁻Gad1⁺* populations and between the two groups from P14 *Sox14^{GFP/+}* data (*Figure 2*), plotted as cumulative proportion of all cells within a given set. The NND distribution is significantly shifted to larger distances between groups than within each of the groups (p<1.4 × 10⁻³⁰, two-sample Kolmogorov–Smirnov test, n = 3 brains). (**B**) Representative z-projections of interneuron distribution amongst TC nuclei, from P14 *Sox14^{GFP/+}* data (*Figure 2*). One dot represents one neuron and they are colour coded by (i) their genetic identity or (ii) spatial cluster. For the spatial clusters a y-projection is also shown. Scale bars, 500 μm. (**C**) Performance of unsupervised k-means algorithm in identifying thalamic interneuron spatial clusters from the P14 *Sox14^{GFP/+}* data (n = 3 brains, see also *Figure 2*) as measured by the silhouette score, which varies with number of clusters (k). We choose k = 2 as this point has the highest score. (**D**) Proportion of *Sox14⁺* and *Sox14⁻* GABAergic cells in each spatial cluster, averaged over three brains (mean ± SEM). The online version of this article includes the following figure supplement(s) for figure 3:

**Figure supplement 1.** Spatial organisation of *Gad1⁺* and *Chrna6⁺* cells in the adult mouse thalamus.

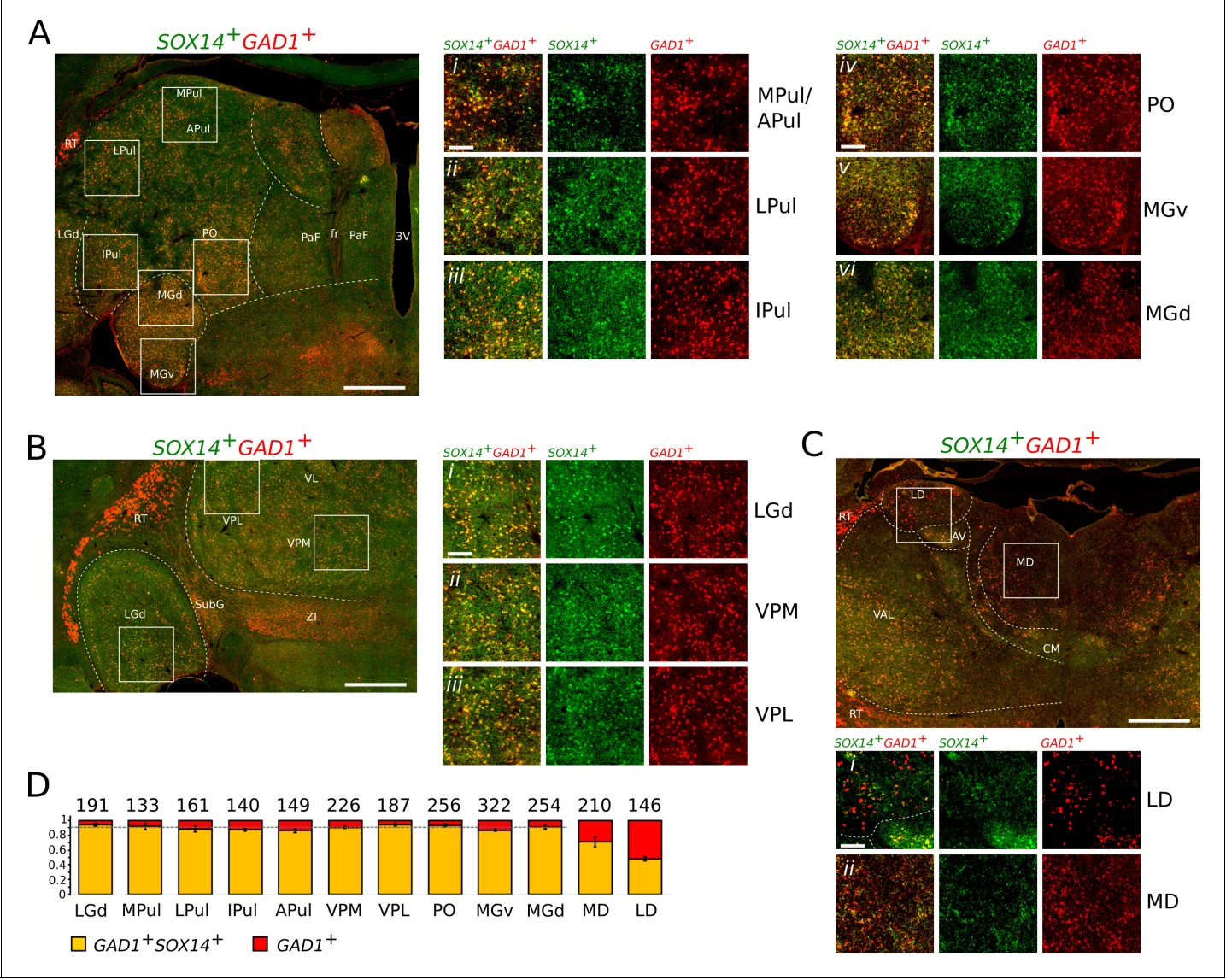

**Figure 4.** *SOX14⁺GAD1⁺* interneurons dominate TC regions of the non-human primate marmoset. Representative coronal sections of the thalamus of a new-born marmoset illustrating the distribution of cells expressing the *SOX14* (green) and *GAD1* (red) mRNAs. (**A**) Caudal plane containing subdivisions of the pulvinar complex, the PO, and the auditory MG. Also visible are parafascicular (PaF) nuclei. Fr, fasciculus retroflexus; 3V, third ventricle. (Ai–iii) Magnifications of indicative areas of the medial (MPul) and anterior pulvinar (APul), the lateral pulvinar (LPul), and inferior pulvinar (IPul). (Aiv). Magnification of a region of the PO. (Av and vi). Magnifications of representative regions of the ventral (MGv) and dorsal (MGd) subdivisions of the auditory thalamus. (**B**) Middle plane section containing the sensory TC nuclei LGd, VPM, VPL, and other non-TC structures (ZI, RT, and SubG). (Bi–iii). Magnifications illustrating the dominant presence of *SOX14⁺GAD1⁺* interneurons in the sensory FO nuclei. (**C**) Anterior plane containing the VAL, centromedial (CM), anteroventral (AV), LD, and MD. The prethalamic RT is recognisable as an entirely *SOX14⁻GAD1⁺* structure. (Ci). Magnification of an area of the LD containing comparable densities of *SOX14⁺* and *SOX14⁻* interneurons. (Cii). Magnification of an area of the MD containing *SOX14⁺* and *SOX14⁻* interneurons. (**D**) Fraction of *SOX14⁺GAD1⁺* (yellow) and *SOX14⁻GAD1⁺* (red) interneurons in selected TC nuclei. Above each bar the total cell counts from nine regions of interest (ROI) measuring 263 µm by 263 µm per each TC nucleus in three age-matched brains (three ROI per TC nucleus per brain). The average fraction of *SOX14⁻GAD1⁺* interneurons deviates significantly from background level in the MD and LD. Scale bars: low magnification; overviews: ~1 mm; magnified areas:~0.2 mm.

and multimodal associative medial, lateral, inferior, and anterior pulvinar subdivisions (MPul, LPul, IPul, and APul, respectively), FO somatosensory VPM and VPL, HO sensory PO, and FO auditory (MGv) and HO auditory (MGd) as well as non-specific HO MD and LD. With the exception of the MD and LD, all TC nuclei tested contained mostly *SOX14⁺GAD1⁺* cells (90.2 ± 0.9%; mean ± SEM; dotted line in *Figure 4D*). This may in fact be an underestimate due to the observed lower efficiency of

the SOX14 probe compared to the GAD1. Analysis of the frequency distribution of the two cell classes across the tested nuclei reveals the MD and LD as outliers of an otherwise normal distribution (*Motulsky and Brown, 2006*). Indeed, in the MD and LD SOX14⁻GAD1⁺ cells account for 28.7 ± 6.7% and 52.3 ± 2.8% (mean ± SEM) of the total GAD1⁺, respectively (*Figure 4D*). The presence of a sizeable population of SOX14⁻GAD1⁺ cells in the MD and LD is intriguing for their reminiscence of the Sox14⁻Gad1⁺ interneurons of the mouse (*Figure 2C,D*) which are also most abundant in these two non-specific HO nuclei.

We find the presence in both the mouse and marmoset of an abundant SOX14⁺GAD1⁺ interneuron class and the relative distribution of SOX14⁺GAD1⁺ and GAD1⁺ single positive interneurons compatible with a conserved basic organisation of interneuron diversity in rodents and primates alike.

## All *Sox14*-expressing thalamic interneurons are born in the midbrain

Given the known role of *Sox14* in specifying subcortical inhibitory classes (*Achim et al., 2013*; *Delogu et al., 2012*; *Guo and Li, 2019*; *Prekop et al., 2018*; *Sellers et al., 2014*) and following our identification of *Sox14* as a conserved genetic marker for the larger cohort of thalamic interneurons in both the primate and rodent brain, we investigated the requirement for this gene in interneurons across TC modalities and hierarchy using the $Sox14^{GFP/GFP}$ (*Sox14* knockout, KO) mouse. We have previously shown that in the Sox14 KO there is a >90% reduction in the number of interneurons in the LGd (*Jager et al., 2016*). We find a comparable reduction in the number of GFP⁺ interneurons overall across the LGd, LP, VP, PO, and MG in the *Sox14* KO (90.5 ± 1.5%, p=2.7 × 10⁻⁴, two-sample two-tailed t-test, n = 3 brains/genotype; *Figure 5A,B*). Conversely, there was no significant change in the number of $Sox14^{-}Gad1^{+}$ cells (p=0.4, two-sample two-tailed t-test; *Figure 5A,B*) and in their distribution across TC regions (*Figure 5A,C*; p>0.05, chi-squared test, n = 3 brains/

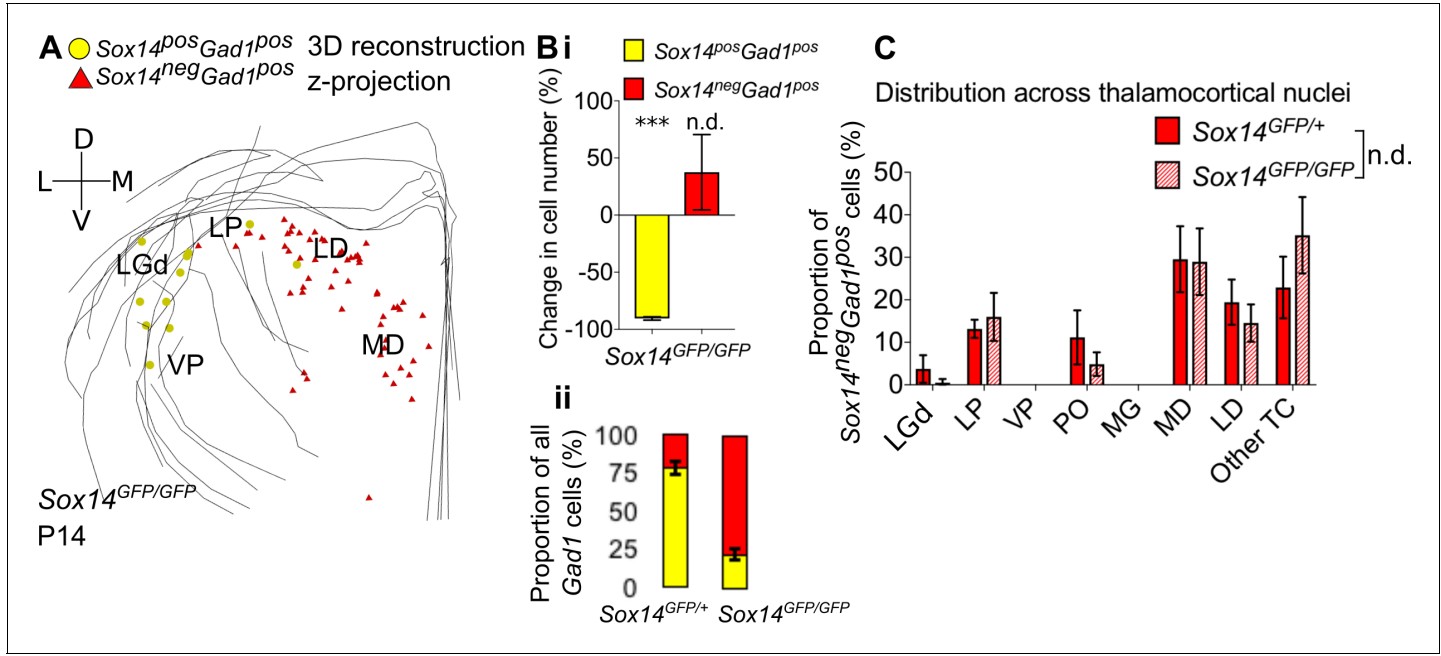

**Figure 5.** Differential requirement for Sox14 highlights two distinct developmental classes. (**A**) Differential requirement for Sox14 highlights two distinct developmental classes. 3D reconstruction of a representative P14 $Sox14^{GFP/GFP}$ thalamus from tracing every tenth 20-µm-thick coronal section, displayed as a z-projection and showing distribution of Sox14⁺Gad1⁺ (yellow) and Sox14⁻Gad1⁺ cells (red). (**B**) (i) Relative change in the number of GFP⁺Gad1⁺ and GFP⁻Gad1⁺ cells across TC regions in P14 $Sox14^{GFP/GFP}$ relative to P14 $Sox14^{GFP/+}$ data (mean ± SEM, n = 3 brains/genotype). There is a significant reduction in the GFP⁺Gad1⁺ population (p=2.7 × 10⁻⁴, two-sample two-tailed t-test), but no statistically significant difference in the size of the GFP⁻Gad1⁺ group (p=0.4, two-sample two-tailed t-test). (ii) Proportion of GFP⁺Gad1⁺ cells within the total GABAergic population is decreased in the $Sox14^{GFP/GFP}$ (mean ± SEM, n = 3 brains/genotype). (**C**) Distribution of GFP⁻Gad1⁺ cells across TC nuclei in the $Sox14^{GFP/+}$ and $Sox14^{GFP/GFP}$ brains at P14 (mean ± SEM; n = 3 brains/genotype). GFP⁻Gad1⁺ distribution is unaltered in the Sox14 KO (p>0.05, chi-squared test).

genotype). These results therefore indicate that the two TC interneuron populations may already be segregated during development and represent two distinct GABAergic lineages.

LGd interneurons in the mouse derive from the midbrain (*Jager et al., 2016*). To explore how the molecular and spatial organisation of thalamic interneurons is generated during development more conclusively, we fate-mapped midbrain lineages and checked for their presence, distribution, and inhibitory profile across the thalamus. We crossed *En1Cre* (*Kimmel et al., 2000*) with a *Rosa26Isl-GFP* (*Sousa et al., 2009*) reporter line (*Figure 6A*; see also *Table 1*), as the *En1* TF gene is expressed in the midbrain and rostral hindbrain progenitors, but not in the forebrain (*Sgaier et al., 2007*). Analysis of the thalamus at P21 reveals GFP+ cells (*En1*+ lineage) distributed across the LGd and co-expressing GABA (*Figure 6B*), therefore independently validating our previous observation (*Jager et al., 2016*). However, like the *Sox14+Gad1+* neurons, *En1*+ cells were observed beyond the LGd – in the LP, VP, PO, and MG, where they were also positive for GABA (*Figure 6B,C*). Plotting their distribution confirmed that it is equivalent to the distribution of *Sox14*+ INs (p>0.05, chi-squared test; *Figure 6C,D*). Occasional GFP+ cells with glia-like morphology were also observed in the thalamus. These cells were GABA- and were not included in any of the analyses.

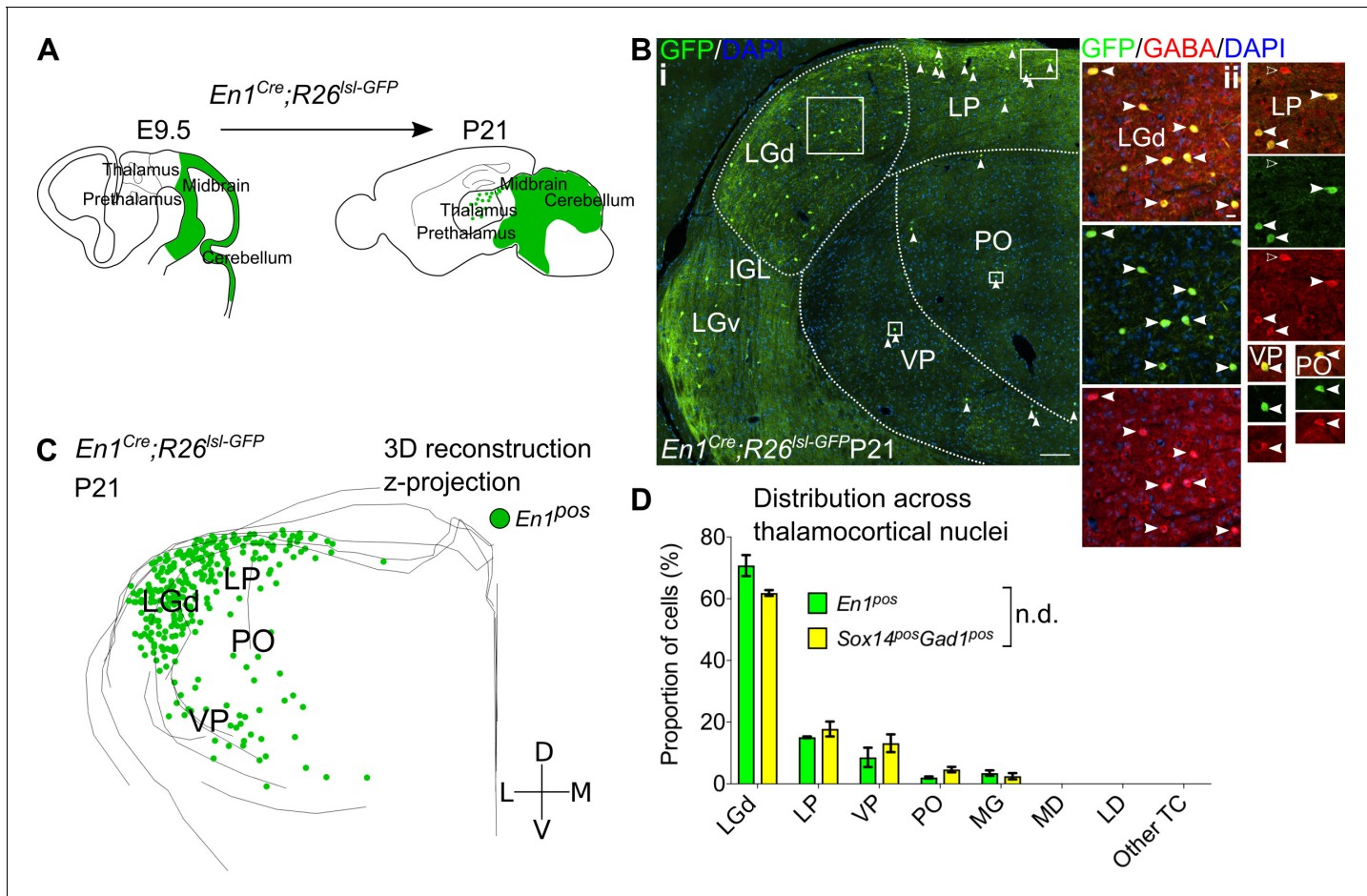

**Figure 6.** *Sox14*+ interneurons in TC regions derive from the midbrain. (**A**) Schematic of the fate mapping experiment: crossing *En1Cre* with *Rosa26Isl-GFP* reporter line permanently labels all midbrain born cells with GFP expression. (**B**) (i) Representative coronal section of P21 *En1Cre; Rosa26Isl-GFP* thalamus with *En1*+ cells observed in the LGd, LP, VP, and PO (considering TC regions only). For clarity some of the *En1*+ cells are indicated with white arrows. Scale bar, 100 μm. (ii) *En1*+ cells in these regions co-express GABA (filled white arrows). Empty arrows mark GABA single-positive cells. Scale bar, 10 μm. (**C**) 3D reconstruction of a representative P21 *En1Cre; Rosa26Isl-GFP* thalamus from tracing every sixth 60-μm-thick coronal section, displayed as a z-projection and showing distribution of *En1*+ cells. (**D**) Distribution of *Sox14+Gad1+* and *En1*+ cells across TC nuclei in *Sox14GFP/+* and *En1Cre; Rosa26Isl-GFP* brains, respectively, plotted as proportion of all the cells within each group (mean ± SEM; n = 3 brains/genotype). The two populations are not differently distributed (p>0.05, chi-squared test).

We therefore conclude that the *Sox14+* thalamic interneurons across FO and HO TC nuclei all derive from the midbrain, and simultaneously that the *Sox14-* GABAergic cells do not; the two classes thus represent distinct inhibitory lineages in TC regions, further supporting their definition as two distinct thalamic interneuron classes.

### Midbrain-derived interneurons migrate along two streams into the sensory thalamus during the first postnatal week

*En1Cre;Rosa26Isl-GFP* line was then used to investigate the timeline and spatial trajectories of the *Sox14+* interneuron precursors migrating from the midbrain into the FO and HO sensory TC regions (*Figure 7A*). Previously, LGd interneurons were found to populate this nucleus in the first postnatal week (*Golding et al., 2014*; *Jager et al., 2016*). We therefore looked at the numbers and migratory morphology of GFP+ (i.e. *En1+*) cells in the thalamus at E16.5, E17.5, P0.5, P1.5, and P2.5. We focused on the LGd, LP, and VP, but left out the PO and MG, due to low overall numbers of interneurons in these two regions (*Figure 1D* and *Figure 2D*).

At E16.5 no GFP+ cells were present in the thalamus. From E17.5 to P2.5 their numbers progressively increased in all of the regions analysed (*Figure 7A,B*). The number of GFP+ cells in the LGd at P2.5 matched previous independent reports (*Golding et al., 2014*), validating our counting method. Midbrain-derived interneurons therefore populate the different TC regions following a similar

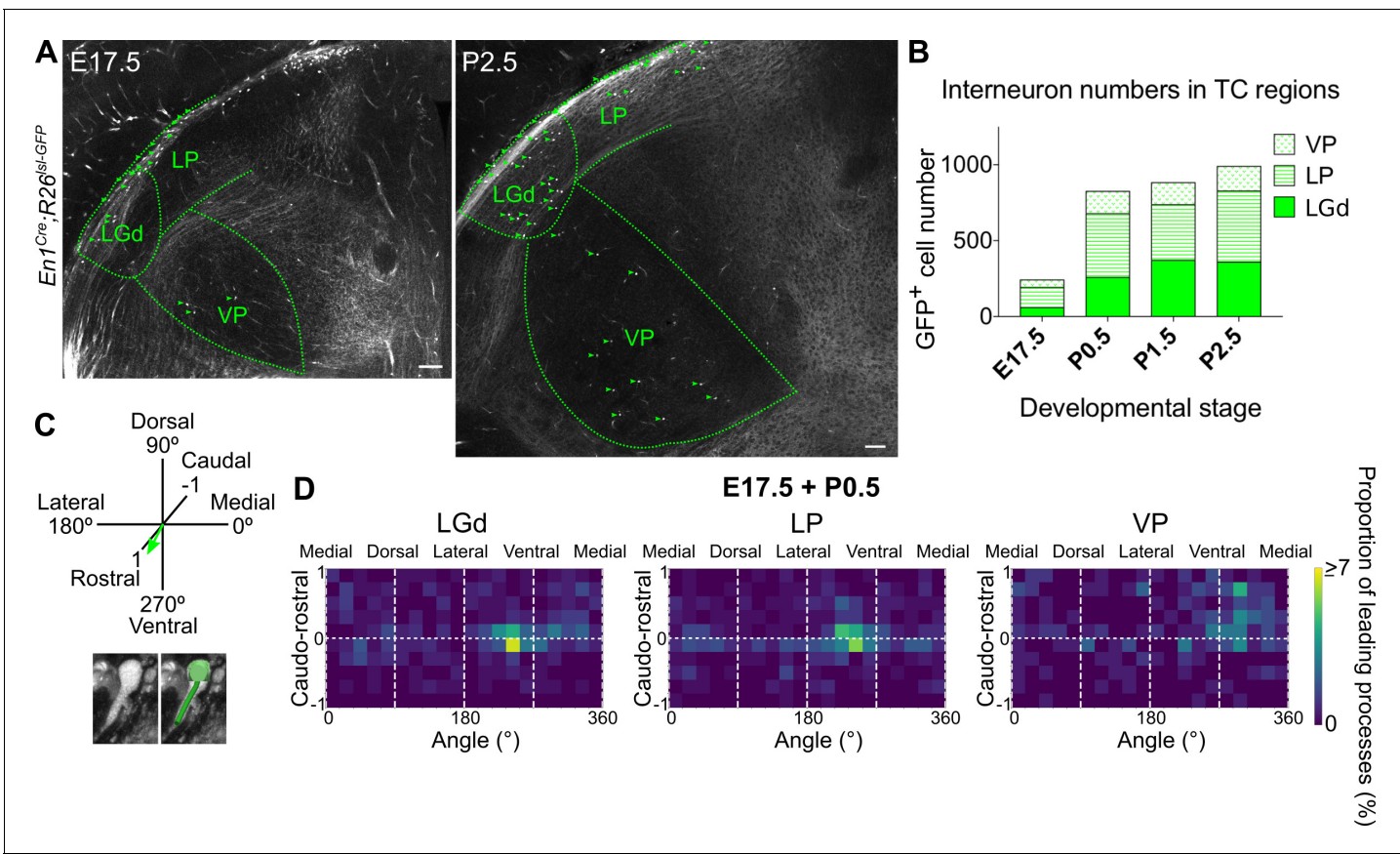

**Figure 7.** Midbrain-derived interneuron precursors progressively populate the thalamus from E17.5 onwards. (**A**) Representative coronal sections of *En1Cre; Rosa26Isl-GFP* thalamus at E17.5 and P2.5. Green arrows mark some of the GFP+ cells. Scale bars, 100 μm. (**B**) Number of GFP+ cells counted in the LGd, LP, and VP from E17.5 to P2.5 (mean, n = 3 brains). (**C**) Leading process orientation of GFP+ cells was determined along the caudo-rostral, ventro-dorsal, and latero-medial dimensions. (**D**) Frequency distribution of leading process orientation for GFP+ cells in the LGd, LP, and VP at E17.5 and P0.5 combined, represented in heat maps (n = 3 brains/developmental stage).

The online version of this article includes the following figure supplement(s) for figure 7:

**Figure supplement 1.** Distinct leading process orientation between interneuron precursors in the LGd and LP compared to VP.

timeline. Interestingly, they appear in two ventrally located nuclei (i.e. LGd and VP) simultaneously (*Figure 7A,B*), implying they use distinct routes to reach them.

To assess quantitatively the direction of migration of the larger, midbrain-derived interneuron class, we determined the leading process orientation of migrating GFP$^+$ cells along all three dimensions: ventro-dorsal, latero-medial, and caudo-rostral; *Figure 7C*; (*Jager et al., 2016*; *Paredes et al., 2016*). This was plotted at a population level as frequency distribution using heat maps, for each nucleus individually, for E17.5 and P0.5 (*Figure 7D*; *Figure 7—figure supplement 1B*), as the relative increase in GFP$^+$ cell numbers was the greatest between these two time points (*Figure 7B*). Moreover, there was a progressive decrease across developmental stages in the proportion of GFP$^+$ cells for which migratory morphology could be identified (*Figure 7—figure supplement 1A*).

Heat maps indicate that at a population level (integrated across dimensions), GFP$^+$ cells migrate into the LGd, LP, and VP in a caudo-rostral and dorso-ventral direction (*Figure 7D*), consistent with the position of the thalamus in the brain relative to their midbrain origin. However, GFP$^+$ precursors in the LGd and LP have a dominant medio-lateral orientation, while those in the VP an opposite, latero-medial orientation, as can also be seen from polar histograms (*Figure 7—figure supplement 1C*). This suggests that midbrain-derived interneuron precursors enter TC regions simultaneously in two distinct streams, one migrating rostro-ventro-laterally to the LGd and LP, and the other rostro-ventro-medially to the VP, indicating a split between visual (LGd, LP) and somatosensory (VP) TC nuclei.

## *Sox14*-negative thalamic interneurons populating HO nuclei are born in the forebrain

With the aim of identifying the origin of the *Sox14*$^-$ interneuron class in the mouse, we first looked for a positive molecular marker that would complement the absence of *Sox14* expression. We made use of DropViz data [available from: dropviz.org; (*Saunders et al., 2018*)] and observed that within inhibitory clusters from the diencephalon, *Sox14* and *Pvalb* show largely non-overlapping expression. It is known that *Pvalb* is expressed by the prethalamic RT (*Clemente-Perez et al., 2017*) and by telencephalic interneuron subtypes derived from the GE (*Marín and Rubenstein, 2001*; *Tasic et al., 2016*; *Tremblay et al., 2016*).

We therefore checked whether any thalamic interneurons are fluorescently labelled using a *Pvalb*$^{Cre}$ mouse (*Hippenmeyer et al., 2005*) crossed to the conditional reporter *Sun1sfGFP* (*Rosa26$^{lsl-nuclear-GFP}$*; see also *Table 1*). These mice contain a CAG promoter driving expression of the mouse nuclear membrane protein SUN1 (Sad1 and UNC84 domain containing 1) fused to two copies of superfolder GFP (sfGFP) inserted in the *Rosa26* locus (*Mo et al., 2015*). Indeed, at P14 GFP was detected at the nuclear membrane of GABA-expressing neurons within the same regions populated by the *Sox14*$^-$ interneurons (*Figure 8—figure supplement 1A*), including the MD and LD, but absent from TC nuclei populated exclusively by *Sox14*$^+$ interneurons, such as the LGd and VP (*Figure 8—figure supplement 1Ai*).

At later ages (P56) *Pvalb* is widely expressed in the mouse thalamus and is observed in high-density gradients in several TC nuclei [2004 Allen Institute for Brain Science. Allen Mouse Brain Atlas. Available from: mouse.brain-map.org; (*Lein et al., 2007*)]. *Pvalb* expression is not conserved across rodents and primates and cannot assist in comparative studies (*Rausell and Jones, 1991*). Importantly, however, in the P14 mouse 93.9% of Pvalb$^+$ cells in TC regions co-expressed GABA (n = 2 brains, *Figure 8—figure supplement 1Aii, B*) and none were labelled by the *En1$^{Cre}$* reporter (0 out of 491 Pvalb$^+$ cells, n = 4 *En1cre; Rosa26$^{lsl-GFP}$* mice) or *Sox14$^{GFP/+}$* reporter(0 out of 206 Pvalb$^+$ cells, n = 2 *Sox14$^{GFP/+}$* mice). Therefore, we define the mouse *Sox14*$^-$ GABAergic cells as *Pvalb*$^+$.

We found the presence of a minor population of Pvalb$^+$ interneurons in the anterior thalamus of the mouse intriguing. Spatial proximity to the RT and shared marker expression (*Pvalb*) may suggest a prethalamic origin for the Sox14$^-$Pvalb$^+$ interneurons. On the other hand, progenitor domains of the telencephalic medial ganglionic eminences (MGE) and preoptic area (POA) also generate *Pvalb*$^+$ interneurons, which are known to integrate in neocortical and hippocampal circuitries (*Gelman et al., 2011*; *Lavdas et al., 1999*; *Wichterle et al., 2001*; *Xu et al., 2004*; *Xu et al., 2008*), but could potentially reach the thalamus. In addition, in humans the *DLX1/2/5*$^+$ GE generate thalamic interneurons selectively for the HO MD and pulvinar nuclei (*Letinić and Kostović, 1997*; *Letinic and Rakic, 2001*; *Rakić and Sidman, 1969*).

We set out first to validate the distinct origin of the Sox14⁻Pvalb⁺ interneurons by fate-mapping rostral forebrain inhibitory progenitor domains using the *Dlx5/6^Cre* (*Monory et al., 2006*) crossed to *Rosa26^lsl-GFP* line (*Figure 8A*; see also *Table 1*). Expression of the distal less homeobox genes *Dlx1/2/5/6* is a common ontogenetic step for all GABAergic lineages in the subpallium, prethalamus, and hypothalamus (*Anderson et al., 1997a*; *Anderson et al., 1997b*; *Bulfone et al., 1993*; *Le et al., 2017*; *Liu et al., 1997*; *Long et al., 2009a*; *Long et al., 2007*; *Long et al., 2009b*; *Simeone et al., 1994*; *Wang et al., 2010*; *Yun et al., 2002*).

At P14 virtually all TC Pvalb⁺ cells are a *Dlx5* lineage (GFP⁺; 99.71 ± 0.29%, n = 3 brains; *Figure 8B,K*) and majority of them co-expressed detectable GABA (93.6 ± 3.7%; *Figure 8B,L*), in line with observations from the *Pvalb^Cre*; *Rosa26^lsl-nuclearGFP* line (*Figure 8—figure supplement 1Aii, B*). We mapped the distribution of Pvalb⁺*Dlx5*⁺ cells across TC regions (*Figure 8C,D*) and observed that it closely recapitulates the distribution of *Sox14⁻Gad1*⁺ cells (*Figure 8C*; p>0.05, chi-squared test).

We also observed other Pvalb⁻*Dlx5*⁺ cells in the thalamus, the majority of which had a glia-like morphology and did not express GABA (*Figure 8—figure supplement 1C*). Occasional Pvalb⁻GABA⁻*Dlx5*⁺ cells with neuronal-like morphology were also seen (*Figure 8—figure supplement 1C,D*), suggesting leaky Cre activity in some cases. That all Pvalb⁺cells in TC nuclei are labelled with GFP argues against this being an artefact of leaky reporting. Pvalb⁻GABA⁻*Dlx5*⁺ cells were not considered in any of the analyses.

While the *Dlx5/6^Cre* fate mapping confirmed the Sox14⁻Pvalb⁺ cells as a distinct, forebrain-derived interneuron in the mouse thalamus, it does not resolve between the different rostral forebrain GABAergic territories of the telencephalon and diencephalon, the latter comprised of the prethalamus and the hypothalamus.

To further discriminate between telencephalic and prethalamic/hypothalamic GABAergic progenitors, we took advantage of the restricted expression of the Forkhead box gene *Foxd1* during the neurogenic period in the prethalamus and hypothalamus (*Newman et al., 2018*; *Shimogori et al., 2010*) and fate-mapped *Foxd1*⁺ lineages by crossing the *Foxd1^Cre* (*Humphreys et al., 2008*) with the *Rosa26^lsl-nuclearGFP* reporter line, described earlier (*Figure 8E*; see also *Table 1*). At P14, GFP was readily detectable at the nuclear membrane of neurons in the prethalamus and, as previously reported (*Newman et al., 2018*), in non-neuronal endothelial cells of the brain's vascular system. Within TC nuclei, colabelling with Pvalb and GABA revealed that the vast majority of thalamic Pvalb⁺ cells derive from *Foxd1*⁺ progenitors (85.56 ± 2.1%; 297 Pvalb⁺ cells from n = 5 *Foxd1^Cre*; *Rosa26^lsl-nuclearGFP* mice) and that virtually all Pvalb⁺*Foxd1*⁺ cells contained detectable GABA (96.43 ± 1.63%). Other GFP nuclei that are neither Pvalb⁺ nor GABA⁺ likely belong to endothelial cells in blood vessels and were NeuN negative (data not shown). The minor residual fraction of Pvalb⁺GFP⁻ cells is similarly distributed across TC nuclei and likely to reflect incomplete *loxp* recombination or very low expression at the *Rosa26* locus.

The fate mapping with *Foxd1^Cre* is therefore consistent with a rostral diencephalic origin of the Sox14⁻Pvalb⁺ interneurons, rather than a telencephalic one, but cannot resolve between prethalamus and hypothalamus. However, *Dlx5*⁺ GABAergic domains of the posterior and ventral hypothalamus can be defined by partially overlapping expression of *Lhx6* or *Nkx2.1* (*Díaz et al., 2014*; *Shimogori et al., 2010*), two transcription factor genes that are also expressed in the GE, but not expressed in the prethalamus (*Figure 8G,I*). Therefore, we tested for a potential hypothalamic origin of the Sox14⁻Pvalb⁺ interneurons by fate mapping GABAergic subdomains of the hypothalamus using the *Nkx2.1^Cre* (*Xu et al., 2008*) and the *Lhx6^Cre* (*Fogarty et al., 2007*) line crossed to the *Rosa26^lsl-GFP* reporter (*Table 1*).

The *Dlx5*⁺*Nkx2.1*⁺ hypothalamic GABAergic progenitors are located in the tuberal and retrotuberal area of the basal hypothalamus (*Díaz et al., 2014*; *Morales-Delgado et al., 2014*; *Shimogori et al., 2010*). Hence, we fate-mapped *Nkx2.1*⁺ lineages with the *Nkx2.1^Cre* line (*Figure 8G*) and investigated the presence of GFP⁺Pvalb⁺ co-expressing neurons in TC regions, at P14. While GFP⁺ cells are present in thalamic territory, none of the TC Pvalb⁺ cells belonged to a *Nkx2.1*⁺ lineage (GFP⁺Pvalb⁺0%, n = 3 brains; *Figure 8Hi*,Hii,K), therefore excluding the posteroventral hypothalamus as a possible source of the *Pvalb*⁺ thalamic interneurons. Importantly, *Nkx2.1* is also a well-established marker of GABAergic progenitors in the MGE and POA (*Shimamura et al., 1995*; *Sussel et al., 1999*) and defines several cortical interneuron lineages including the fast spiking Pvalb⁺ interneurons, the somatostatin⁺ interneurons (*Fogarty et al., 2007*; *Pleasure et al., 2000*;

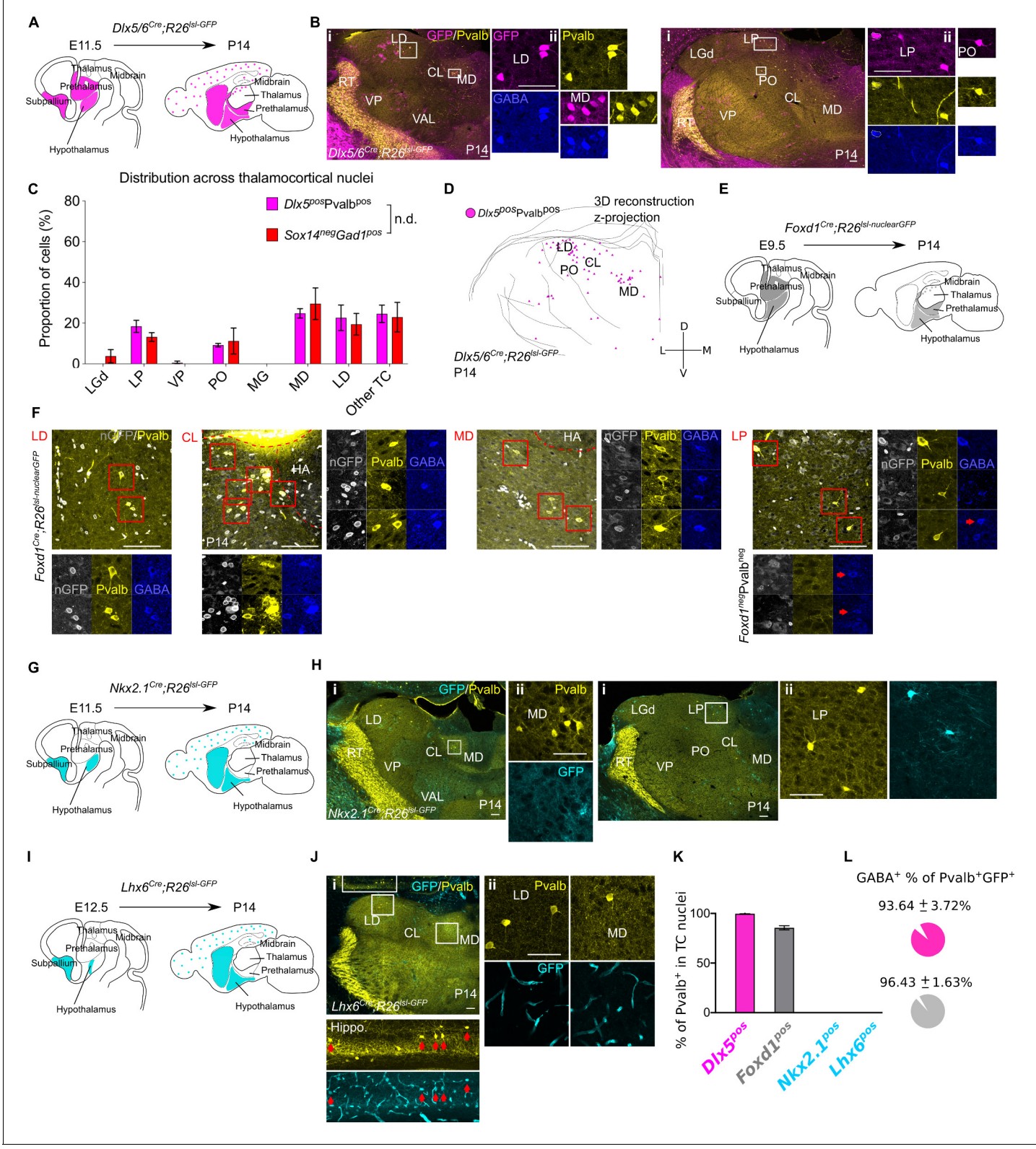

**Figure 8.** *Sox14⁻Pvalb⁺* interneurons in TC regions derive from the *Dlx5ᵖᵒˢ*, *Foxd1ᵖᵒˢ*, *Nkx2.1ⁿᵉᵍ*, and *Lhx6ⁿᵉᵍ* rostral forebrain. (**A**) Schematic of the fate mapping experiment: crossing *Dlx5/6ᶜʳᵉ* with *Rosa26ˡˢˡ⁻ᴳᶠᴾ* reporter line permanently labels all ventral telencephalic, hypothalamic, and prethalamic-born cells with GFP expression. (**B**) (**i**) Representative coronal sections of P14 *Dlx5/6ᶜʳᵉ; Rosa26ˡˢˡ⁻ᴳᶠᴾ* thalamus with *Dlx5⁺*Pvalb⁺ cells present in the MD, LD, CL, VAL, VM, LP, and PO (considering TC regions only). Scale bar, 100 μm. (**ii**) *Dlx5⁺*Pvalb⁺ cells in TC regions co-express GABA. Scale bar, 100 μm.
*Figure 8 continued on next page*

Figure 8 continued

(C) Distribution of $Dlx5^+$Pvalb$^+$ and $Sox14^-$Gad1$^+$ cells across TC nuclei in P14 $Dlx5/6^{Cre}$; $Rosa26^{lsl-GFP}$ and $Sox14^{GFP/+}$ brains, respectively, plotted as proportion of all the cells within each group (mean ± SEM, n = 3 brains/genotype). The two populations are not differently distributed (p>0.05, chi-squared test). (D) 3D reconstruction of a representative P14 $Dlx5/6^{Cre}$; $Rosa26^{lsl-GFP}$ thalamus from tracing every sixth 60-μm-thick coronal section, displayed as a z-projection and showing distribution of $Dlx5^+$Pvalb$^+$ cells. (E) Schematic of the fate mapping experiment crossing the $Foxd1^{Cre}$ with $Rosa26^{lsl-nuclearGFP}$ reporter line to permanently label hypothalamic and prethalamic-born cells with nuclear membrane localised GFP. (F) Representative coronal sections of P14 $Foxd1^{Cre}$; $Rosa26^{lsl-nuclearGFP}$ thalamus with $Foxd1^+$Pvalb$^+$ cells present in the LD, CL, MD, and LP. Scale bar, 100 μm. Enlarged areas (red boxes) showing $Foxd1^+$Pvalb$^+$ cells co-expressing GABA. Additional GABA$^+$ but $Foxd1^-$Pvalb$^-$ neurons are also visible in the LP (red arrows). HA: habenula. (G) Schematic of the fate mapping experiment: crossing $Nkx2.1^{Cre}$ with $Rosa26^{lsl-GFP}$ reporter line permanently labels some hypothalamic and all MGE-born cells with GFP expression. (H) (i) Representative coronal sections of P14 $Nkx2.1^{Cre}$; $Rosa26^{lsl-GFP}$ thalamus with Pvalb$^+$ and $Nkx2.1^+$ cells present in the MD, LD, CL, VAL, VM, LP, and PO (considering TC regions only). Scale bar, 100 μm. (ii) $Nkx2.1^+$ cells in TC regions do not co-express Pvalb$^+$. Scale bar, 100 μm. (I).Schematic of the fate mapping experiment crossing the $Lhx6^{Cre}$ with $Rosa26^{lsl-GFP}$ reporter line to permanently label some hypothalamic and MGE born cells with GFP. (J) (i) Representative coronal section of P14 $Lhx6^{Cre}$; $Rosa26^{lsl-GFP}$ showing GFP$^+$Pvalb$^+$ cells in the hippocampus (red arrows) and GFP$^-$Pvalb$^+$ present in the MD, LD, and CL nuclei of the thalamus (considering TC regions only). Scale bar, 100 μm. (ii) GFP$^+$ cells in TC regions are endothelial and do not co-express Pvalb$^+$. Scale bar, 100 μm. (K) Proportion of Pvalb$^+$ cells in TC regions that are $Dlx5^+$, $Foxd1^+$, $Nkx2.1^+$, or $Lhx6^+$ at P14 (mean ± SEM, $Dlx5^+$ n = 3 brains, $Foxd1^+$ n = 5 brains, $Nkx2.1^+$ n = 3 brains, and $Lhx6^+$ n = 4 brains). (L) Proportion of $Dlx5^+$Pvalb$^+$ and $Foxd1^+$Pvalb$^+$ cells in TC regions co-expressing GABA at P14 (mean ± SEM, $Dlx5^+$ n = 3 brains, $Foxd1^+$ n = 5 brains).
The online version of this article includes the following figure supplement(s) for figure 8:

**Figure supplement 1.** Pvalb is a marker for $Sox14^-$ thalamic interneurons.

Xu et al., 2004; Xu et al., 2008), and from the POA, Npy$^+$, or Reelin$^+$ interneurons (Gelman et al., 2011; Gelman et al., 2009). Hence this result strengthens the evidence against a telencephalic contribution to thalamic interneurons in the mouse.

Although we did not conduct a detailed investigation of Pvalb$^-$ cell types labelled by the $Nkx2.1^{Cre}$; $Rosa26^{lsl-GFP}$ reporter in the thalamus, we noted several glia-like morphologies that were also negative for the pan-neuronal marker NeuN (data not shown).

The hypothalamic domain of $Lhx6$ expression only partially overlaps with that one of $Nkx2.1$ and defines additional $Dlx5^+$ GABAergic progenitors of the hypothalamus (Díaz et al., 2014; Kim et al., 2020; Shimogori et al., 2010). Hence, we fate-mapped $Lhx6^+$ lineages with the $Lhx6^{Cre}$ line (Figure 8I) and investigated the presence of GFP$^+$Pvalb$^+$ co-expressing neurons in TC regions, at P14. As no GFP$^+$ cells were present in thalamic territory (GFP$^+$Pvalb$^+$0%, n = 4 brains; Figure 8Ji,Jii, K), we excluded the alar hypothalamus as a potential source of the Pvalb$^+$ thalamic interneurons. Importantly, $Lhx6$ marks also the vast majority of GE-derived telencephalic interneurons (Zhao et al., 2008), hence this result further confirms that the telencephalic GE are an unlikely source of thalamic interneurons in the mouse.

Altogether, we therefore conclude that the rarer $Sox14^-$ thalamic interneuron class is a distinct lineage compared to the larger, midbrain-born $Sox14^+$ thalamic interneuron class, and that originates from $Dlx5^+Foxd1^+$ progenitors in the prethalamus.

## Discussion

Our study reveals a previously unappreciated complexity of GABAergic interneurons in the mouse TC nuclei, demonstrating that interneurons are not restricted to the FO visual thalamus, but present across modalities and hierarchical levels, including limbic structures.

We recognise two broad thalamic interneuron classes, defined by their origin in either the $En1^+$ midbrain or the $Dlx5^+Foxd1^+Nkx2.1^-Lhx6^-$ rostral forebrain. However, the two ontogenetic programmes contribute differentially to interneuron numbers, with the midbrain-derived class overwhelmingly more abundant.

The midbrain-derived interneurons depend on the Sox14 transcription factor, a gene that we had previously implicated in LGd interneuron differentiation (Jager et al., 2016) and a known postmitotic marker for GABAergic subtype neurogenesis in the brainstem (Achim et al., 2013; Achim et al., 2014; Huisman et al., 2019; Prekop et al., 2018). Taking advantage of a $Sox14^{GFP}$ mouse line, we now provide absolute numbers and standardised anatomical distribution of this major class of interneurons across the entire thalamus, in the $Sox14^{GFP/+}$ C57Bl/6 genetic background.

Rather than representing a peculiarity of mice, the midbrain ontogenetic programme may be the dominant source of thalamic interneurons in larger-brained mammals, as suggested by the

identification of $SOX14^+GAD1^+$ interneurons across virtually all TC nuclei in the interneuron-rich thalamus of the marmoset. This finding is consistent with the recent report of molecularly defined classes of GABAergic interneurons in the macaque and human thalamus, where $SOX14$ expression differentiates between interneuron subclasses. Interestingly, $SOX14^-$ interneurons are $DLX1-6^+$, in agreement with the ontogenetic plan that we propose here (*Bakken et al., 2020*). Consistent with a conserved midbrain ontogeny of thalamic interneurons, Jones previously described late appearance of interneurons in the ferret and macaque thalamus, progressively from caudal towards rostral nuclei (*Hayes et al., 2003*; *Jones, 2002*). It can also be seen from the BrainSpan Atlas of the Developing Human Brain [BrainSpan Atlas of the Developing Human Brain; available from: http://www.brainspan.org; (*Miller et al., 2014*)] that both $GAD1$ and $SOX14$ expression increase in the dorsal thalamus in the mid-prenatal period (from postconception week 16), which is also consistent with a migration of midbrain-born interneurons into these regions.

Interestingly, grafting experiments using chick and quail embryos demonstrated a potential for midbrain cells to populate retino-recipient nuclei in the chick diencephalon (*Martinez and Alvarado-Mallart, 1989*). The grafted midbrain cells were observed migrating tangentially at the surface of the diencephalon and seemingly through the host optic tract before invading the regions targeted by the retinal projections (*Martinez and Alvarado-Mallart, 1989*). The neurotransmitter identity of these migrating cells is unknown, but their midbrain origin and distribution across the thalamus resemble the mouse $Sox14^+$ interneurons, suggesting that in birds too, the largest cohort of interneurons is a midbrain lineage. Relatedly, lineage tracing in chick, using a retroviral library, indicated that clonally related siblings can populate both the diencephalon and mesencephalon (*Golden and Cepko, 1996*), in keeping with a revised model of evolutionary relationship of caudal diencephalon and midbrain (*Albuixech-Crespo et al., 2017*). The distribution of $Sox14^+$ interneurons observed in the mouse is similar to the one described in the opossum, a living marsupial thought to resemble early mammals in the fossil record (*Penny et al., 1984*). Intriguingly, rather than spreading throughout the nucleus, interneurons occupy the lateral subdivision of the LP in the adult opossum thalamus (*Penny et al., 1984*), reminiscent of the route taken by migrating midbrain-derived interneuron precursors in the developing mouse thalamus.

While the emerging picture points to a midbrain ontogeny for the largest fraction of thalamic interneurons, this is not sufficient to explain the overall developmental complexity of interneurons in the thalamus. In both the mouse and marmoset, we now report the presence of $Sox14^-Gad1^+$ interneurons with drastically more restricted distribution. In the mouse this interneuron class is found enriched in HO TC nuclei (the non-specific MD and LD, but also sensory-related LP and PO). Similarly, in the marmoset $SOX14^-GAD1^+$ interneurons are also a minor class enriched in the HO nuclei MD and LD in the anterior portion of the thalamus. This specific distribution is intriguing as it may reflect the requirement in some associative nuclei for interneurons with unique functional properties that the larger midbrain-derived class cannot provide. Such hypothetical evolutionary drive is not dissimilar to the one previously proposed for some interneurons of the human thalamus, where GE-derived $DLX1/2^+$ interneurons were shown to migrate into associative nuclei MD and pulvinar (*Letinić and Kostović, 1997*; *Letinic and Rakic, 2001*; *Rakić and Sidman, 1969*).

While a dual midbrain and forebrain ontogeny of thalamic interneurons emerges as the conserved mammalian blueprint for thalamic interneuron organisation, with the midbrain-derived class contributing the largest proportion of interneurons and the forebrain-derived class enriched in selected HO TC nuclei, species-specific differences also exist. In the mouse, midbrain- and forebrain-derived interneurons are spatially segregated along clear anatomical and functional subdivisions of the thalamus, in the marmoset, presumptive midbrain-derived interneurons expanded dramatically and are more broadly distributed. The forebrain-derived interneurons are not found in precisely the same set of associative TC nuclei across mouse and marmoset, while the forebrain-derived human lineage previously described is thought to be an evolutionary innovation that migrates into the thalamus from the GE along transient anatomical structures that are not present in rodents and non-human primates (*Letinić and Kostović, 1997*). Consistent with this, our data support a model whereby the mouse, and by extension the marmoset $Sox14^-Gad1^+$ interneurons are specified in the $Nkx2.1^-Lhx6^-Dlx5^+Foxd1^+$ prethalamus, rather than subpallium. Technical limitations make a detailed assessment of lineage descent in non-human primates and humans more challenging and whether humans retained the prethalamic interneuron class that we described here is currently unknown.

Further investigation of species-specific differences may provide important cues to trace the evolution of the mammalian TC system, using interneurons as the key to unravel its complexity.

## Materials and methods

### Animals

The mice were housed in the animal facilities at King's College London under standard conditions on a 12 hr:12 hr dark/light cycle, with unrestricted access to water and food. Housing and experimental procedures were approved by the King's College London Ethical Committee and conformed to the regulations of the UK Home Office personal and project licences under the UK Animals (Scientific Procedures) 1986 Act. Both female and male mice were used in a randomised way across experiments. The morning when the vaginal plug was observed was designated as embryonic day (E) 0.5 and the day of birth as postnatal day (P) 0.5 (see also *Table 1*).

### Callithrix jacchus

A total of seven New World marmoset (*C. jacchus*) monkeys were used in this study. All experiments were conducted in accordance with the guidelines approved by the RIKEN Institutional Animal Care (W2020-2-022).

### Immunohistochemistry and ISH

Mice were transcardially perfused with 4% PFA and the brains dissected and postfixed in PFA at 4°C overnight, then washed in PBS for at least 24 hr at 4°C. For ISH, brains were stored in PFA for 5 days, to minimise RNA degradation, and all the subsequent solutions were treated with diethyl pyrocarbonate (DEPC; AppliChem). The brains were cryoprotected in a sucrose gradient (10–20–30%), frozen on dry ice, and cryosectioned as 20 μm coronal sections collected on Superfrost Ultra Plus slides (Thermo Scientific) for ISH, or as 60 μm free-floating coronal sections for IHC.

### Immunohistochemistry

Brain sections were washed in PBS three times and blocked in 2–7% normal goat serum (NGS) solution (in 1× PBS, 0.1–0.3% Triton-X100) for 2 hr at room temperature (RT). Primary antibodies (*Table 2*) were diluted in blocking solution and incubated with the sections (as stated in the table). This was followed by three 30 min PBS washes, and incubation in secondary antibodies (*Table 2*) diluted 1:500 in blocking solution, for 2 hr at RT. After two 30 min PBS washes, the sections were incubated in DAPI for 30 min (1:40,000 dilution in PBS; Life Technologies) and mounted using Pro-Long Gold mounting media (Invitrogen).

### In situ hybridisation

*Gad1* antisense RNA probe was transcribed in vitro from full-length cDNA template (IMAGE ID: 5358787). The probe was diluted to a final concentration of 800 ng/ml in hybridisation buffer (50% formamide, 10% dextran sulphate, 1 mg/ml rRNA, 1× Denhardt's solution, 0.2 M NaCl, 10 mM Tris HCl, 5 mM $NaH_2PO_4.2H_2O$, 1 mM Tris base, 50 mM EDTA) and applied onto the slides, which were

**Table 2.** Antibodies.

| Antibody | Dilution | Incubation time | Source |
|---|---|---|---|
| Rabbit anti-GABA | 1:2000 | 2X ON, 4°C | Sigma, A2052 |
| Chicken anti-Gfp | 1:5000 | 2X ON, 4°C | Abcam, Ab13970 |
| Mouse anti-parvalbumin | 1:2000 | 1X ON, 4°C | Sigma-Aldrich, P3088 |
| Goat anti-chicken Alexa-488 | 1:500 | 2 hr, RT | Invitrogen, A11039 |
| Goat anti-rabbit Alexa-568 | 1:500 | 2 hr, RT | Invitrogen, A11036 |
| Goat anti-rabbit Alexa-647 | 1:500 | 2 hr, RT | Invitrogen, A21245 |
| Goat anti-mouse Alexa-568 | 1:500 | 2 hr, RT | Invitrogen, A11004 |
| Goat anti-mouse Alexa-635 | 1:500 | 2 hr, RT | Invitrogen, A31575 |

incubated in a humidified chamber at 65℃ overnight. The slides were then washed three times for 30 min in wash buffer (50% formamide, 1× SSC, 0.1% Tween) at 65℃, two times for 30 min in MABT buffer (100 mM maleic acid, 150 mM NaCl, 0.1% Tween-20) at RT, and blocked for 2 hr at RT (2% Boehringer Blocking Reagent (Roche), 20% inactivated sheep serum in MABT). Sheep a-DIG alkaline phosphatase conjugated antibody (Roche, 11093274910) was diluted 1:2000 in the blocking solution and incubated with the slides overnight at 4℃. This was followed by five 20 min washes in MABT and two 20 min washes in the AP buffer (0.1M Tris-HCl pH 8.2, 0.1%-Tween-20). Fast red TR/Naphthol AS-MX tablets (Sigma) were dissolved in the AP buffer and applied onto the slides for colour reaction for 3–6 hr at RT in the dark. The slides were then washed three times for 20 min in PBS before proceeding with IHC for GFP as described above. $Sox14^{GFP/+}$ and $Sox14^{GFP/GFP}$ sections were always processed in parallel.

### ISH in *C. jacchus*

Fluorescent ISH was performed as previously described (*Watakabe et al., 2006*) with some modifications. Riboprobes incorporating digoxigenin (DIG) and fluorescein (FL) were hybridised overnight. After washing, FL- and DIG-labelled probes were each detected in different ways. For detection of the DIG probes, the sections were incubated with an anti-DIG antibody conjugated with horse radish peroxidase (HRP) (1/500, Roche Diagnostics) for 6 hr at room temperature. After washing in TNTx (0.1 M Tris-HCl, pH 7.5, 0.15 M NaCl, 0.05% Triton X-100) three times for 5 min, the sections were treated with 1:100 diluted TSA-Plus (DNP) reagents (Perkin Elmer) for 20 min. After washing in TNTx 3 × 10 min, the sections were incubated for 2 hr at room temperature with an anti-DNP antibody conjugated with Alexa 488 (1/500, Invitrogen). After quenching HRP activity and washing, the sections were incubated for 2 hr at room temperature with an anti-FL antibody conjugated with HRP (1/500, Roche Diagnostics) followed by reaction with TSA biotin reagents (Perkin Elmer) and visualisation with streptavidin conjugated with Alexa594 (Invitrogen).

### Quantifying distribution of neuronal populations in histological sections

#### In mice

Confocal z-stacks covering the extent of the thalamus across all axes (caudo-rostral, ventro-dorsal, and latero-medial) were acquired using either Nikon A1R inverted confocal, inverted spinning disk Nikon Ti microscope or Olympus VS120 slide scanner, with 10× (NA 0.30 Plan Fluor DLL) and 20× (NA 0.75 Plan Apo VC or UPLSAPO NA 0.75) objectives. The stacks were then viewed with the Neurolucida software. TC nuclei were identified from the DAPI counterstain, using cytoarchitectonically recognisable structures, such as the LGd, the habenular complex, the RT, the anterior pretectum, and the fasciculus retroflexus (fr), as landmarks for orientation and reference. When the nuclear GFP reporter was used, only cells with detectable DAPI signal were included in the calculations. The cells of interest (*Table 3*) were assigned to TC regions by comparing the sections to the Allen Brain Reference Atlas and annotated and counted manually. For each brain, only one hemisphere was analysed (chosen in a randomised way). For experiments using $Gad1^+$ and $Chrna6^+$ ISH data from the Allen Mouse Brain Atlas resource [2004 Allen Institute for Brain Science. Allen Mouse Brain Atlas. Available from: mouse.brain-map.org; (*Lein et al., 2007*)], all images of P56 C57BL/6J coronal brain sections containing the thalamus were downloaded for each gene (every eighth 25-µm-thick section, sampling every 200 µm across the thalamus), and analysed in the same way as described above.

#### In marmoset

Images were acquired with a fluorescence microscope BZ-X810 (Keyence) or BZ-9000 (Keyence). Representative coronal sections at anterior, intermediate, and posterior levels were analysed manually, by delineating nuclear boundaries according to the neonate Marmoset Gene Atlas, RIKEN CBS, Japan (https://gene-atlas.brainminds.riken.jp). Cell counting was conducted using the Cell Counter Plugin and ROI manager in ImageJ (*Schindelin et al., 2015*). Within the boundaries of each TC nucleus analysed, three ROIs of 263 µm by 263 µm were positioned randomly and their content of single positive or double positive cells added together to generate a representative fraction of $GAD1^+$ and $GAD1^+SOX14^+$ cells for the TC nucleus (no SOX14 single positive cells were detected). Counts were replicated in three age matched brains to calculate mean ± SEM.

**Table 3.** Genetic identity of cells counted across TC regions and technical details of corresponding experiments.

| Transgenic line | Age | Cells annotated/ counted | Number of brains | Sampling | Section thickness (µm) |
|---|---|---|---|---|---|
| **Mouse** | | | | | |
| Sox14$^{GFP/+}$ | P21 | GFP$^+$ | 3 | Whole thalamus | 50 (10; optical sections) |
| Sox14$^{GFP/+}$ | P14 | GFP$^+$ and Gad1$^+$ | 3 | Every tenth coronal section | 20 |
| Sox14$^{GFP/+}$ | P14 | GFP$^+$ and Pvalb$^+$ | 2 | Every sixth coronal section | 60 |
| Sox14$^{GFP/GFP}$ | P14 | GFP$^+$ and Gad1$^+$ | 3 | Every tenth coronal section | 20 |
| En1$^{Cre}$; Rosa26$^{lsl-GFP}$ | P21-30 | GFP$^+$ | 3 | Every sixth coronal section | 60 |
| En1$^{Cre}$; Rosa26$^{lsl-GFP}$ | P14 | GFP$^+$ and Pvalb$^+$ | 3 | Every sixth coronal section | 60 |
| Dlx5/6$^{Cre}$; Rosa26$^{lsl-GFP}$ | P14 | GFP$^+$, Pvalb$^+$, and GABA$^+$ | 3 | Every sixth coronal section | 60 |
| Foxd1$^{Cre}$; Rosa26$^{lsl-nuclearGFP}$ | P14 | GFP$^+$, Pvalb$^+$, and GABA$^+$ | 5 | Every sixth coronal section | 30 |
| Nkx2.1$^{Cre}$; Rosa26$^{lsl-GFP}$ | P14 | GFP$^+$ and Pvalb$^+$ | 3 | Every sixth coronal section | 60 |
| Lhx6$^{Cre}$; Rosa26$^{lsl-GFP}$ | P14 | GFP$^+$ and Pvalb$^+$ | 4 | Every sixth coronal section | 60 |
| Pvalb$^{Cre}$; Rosa26$^{lsl-nuclearGFP}$ | P14 | GFP$^+$, Pvalb$^+$, and GABA$^+$ | 2 | Every sixth coronal section | 60 |
| **Marmoset** | | | | | |
| Wild type | P0 | SOX14 and GAD1 | 3 | Representative anterior, intermediate and posterior planes | 28 |

## 3D reconstructions of cell distributions from histological sections

3D reconstructions of cell distributions (*Table 3*) across thalamic regions were generated for each brain separately using the Neurolucida software (MBF Bioscience), from the acquired confocal z-stacks or Allen Mouse Brain Atlas ISH data as described above. For each image the outline of the thalamus and the surrounding structures were manually traced using the 'contour' function and the cells were annotated with the 'marker' function, placed at the centre of the soma. Traced images were then aligned in sequential rostro-caudal order, manually for each brain, using tissue landmarks (midline and clearly recognisable structures, for example, LGd, RT, habenula, hippocampus) for reference, and their spacing in the rostro-caudal dimension was preserved according to the sampling used for each brain.

## 3D reconstructions of cell distributions by whole brain serial two photon imaging

Sox14$^{GFP/+}$ mouse brain samples (P21; n = 3) were embedded in a 4.5% oxidised-agarose solution containing agarose (type 1; Sigma), 10 mM NaIO4 (Sigma) and 50 mM phosphate buffer (PB). Samples were imaged with TissueCyte 1000 (*Ragan et al., 2012*) with a custom cooling system (JULABO UK Ltd.) for serial two-photon (STP) tomography across the complete mouse brain. Physical sectioning was performed every 50 µm with optical sectioning every 10 µm. A 16×, 0.8 NA immersion objective (Nikon Inc) acquired 1 × 1 mm image tiles at spatial resolution 0.54 µm with a 12 × 10 tiling mosaic required to obtain a complete coronal tissue section. Laser (Chameleon Ultra II, Coherent) excitation was conducted at 920 nm for GFP excitation with three PMT channel acquisition for red, green, and blue wavelength collection. STP imaging occurred over 5 days and generated 3.5 terabytes of data per brain. Tiled data was stitched alongside STP acquisition using a custom Python and ImageJ/Fiji pipeline.

STP data sets of each mouse brain were down-sampled to 10 µm isotropic voxel size and registered with the Allen CCF3 average atlas using Elastix (*Klein et al., 2010*). Registration was performed from average atlas (moving) to down-sampled STP (fixed) using a combination of rigid, affine and b-spline transformation steps, executed using a multiresolution approach for robust global and local structure registration. An advanced Mattes Mutual Information similarity metric and an

adaptive stochastic gradient descent objective optimiser were used at each transformation step with the transformation at each step combined into a final transformation map which was applied to the CCF3 annotation atlas and a custom hemisphere atlas used to distinguish structures across hemisphere. Registration resulted in a spatial mapping from the STP data to the CCF3 atlas space allowing the delineation of thousands of anatomical structures according to the Allen Brain Atlas hierarchically organised taxonomy.

For automated cell counting, a U-Net (Ronneberger O., 2015) deep learning network was trained to segment fluorescently labelled cells in STP and confocal data sets. During training, 219 images of fluorescently labelled cells (512 × 512 pixels; 0.54 µm voxel size) were manually segmented using ImageJ/Fiji. Images came from STP and confocal data sets of GFP labelled cells from transgenic mouse lines and viral tracing studies and contained GFP expression localised to soma and dendritic and axonal structures. During manual segmentation, only soma localised fluorescence was labelled. To increase generalisation of the network for robust performance on new data, drop out layers at 50% probability were introduced into the network, plus image augmentation was used to increase the initial 219 image data set by 56-fold. Augmentation operations included image flipping, rotation in the range −360° to +360°, zooming in the range 90–110%, skewing, a random elastic distortion using a grid size of 10 pixel spacing, shearing and a custom Poisson noise addition. Some transformations were assisted using the Python package Augmentor (*D Bloice et al., 2017*), with the custom Poisson noise generation written as a class to interface with the Augmentor package. Each transformation was given a 50% probability of occurring and resulted in a final training data set of 12,264 image and annotated pairs. Training data was split 75% (9198 samples) for training the network and 25% (3066 samples) for validating the network with conscious effort to maintain class balance between the training and validation to prevent overfitting or loss issues during training.

The model was trained with the ELU activation function, using an Adam optimiser with a Binary Cross-entropy loss function. A batch size of 8 was used with early stopping evoked if a validation dice loss score did not improve after 30 epochs of training. Model training was performed on a workstation equipped with a NVIDIA Titan Xp GPU using Python and the TensorFlow 2.0 platform.

For automated thalamus counting, all structures belonging to the thalamus, according to the Allen Brain Atlas hierarchically organised taxonomy, were extracted from the registered STP data sets using masks upsampled to the original 0.54 µm data and fed into the trained network for automated segmentation. Correction for oversampling of cells in the axial axis was done by grouping detected cells if they overlapped within a radius of 10 µm, and subsequently keeping the centrally positioned cell in the axial axis. Automated counting in the entire thalamus took 7 hr per sample using an Ubuntu Intel(R) Core(TM) i9-7980XE CPU at 2.60 GHz workstation, with 32 cores and 128 GB RAM.

## NND calculations

Nearest neighbour distance (NND) was determined for the *Sox14+Gad1+* and *Sox14-Gad1+* cells from the 3D reconstructions of their distributions. The cells' coordinates in 3D were generated by Neurolucida and analysed using a custom Python script and the Pandas library (*McKinney et al., 2010*) to calculate NNDs separately for each group and between the two groups, for each *Sox14GFP/+* brain individually. The data was then normalised to the largest NND within each data set (each individual group and between groups sets for each brain) averaged across the brains (mean ± SEM) and plotted as cumulative distribution. Normalisation allows us to plot their cumulative distribution as a fraction of the maximum distance, though even before normalisation of the curves were broadly similar. Statistically significant differences between the distributions were verified using the two-sample Kolmogorov–Smirnov test, implemented in the SciPy library (*Jones et al., 2001*).

## Migratory morphology analysis

E16.5, E17.5, P0.5, P1.5 (n = 3 brains/developmental stage), and P2.5 (n = 1) *En1Cre; Rosa26lsl-GFP* brains were quickly dissected on ice and immersed in 4% PFA for 12 hr before switching to PBS. 300-µm-thick coronal sections were cut on a vibratome (Leica VT 1200S). To increase the imaging depth, the sections were cleared following the ScaleSQ protocol (*Hama et al., 2015*). ScaleS4 buffer was used as a mounting medium (*Hama et al., 2015*), and spacers were placed on the slides to prevent compressing the sections. Nikon A1R inverted confocal was used to acquire z-stacks that

covered the entire extent of the thalamus for each brain, with a 20× objective (NA 0.75 Plan Apo VC). The achieved imaging depth in *z* ranged from 200 to 250 μm. The stacks were imported into Neurolucida software (MBF Bioscience) to trace the migratory morphology of GFP$^+$ cells in the LGd, LP, and VP. On average, two sections covered the extent of these nuclei in the rostro-caudal dimension and the first time point when GFP$^+$ cells were observed there was at E17.5. GFP$^+$ cells were not traced in the PO and MG due to their low numbers in these nuclei in the juvenile and adult brains, and the ambiguity in delineating these regions anatomically in the embryonic brains. We did not observe GFP$^+$ cells with neuronal morphology in any other TC regions (i.e. outside the FO and HO sensory thalamus) for all ages analysed. In the analysed regions (LGd, LP, and VP), all GFP$^+$ somas were annotated using the semi-automated 'Soma' function. The leading processes were traced manually with the 'Tree' function, starting in the middle of the soma and until each process could be unequivocally identified or until the point of bifurcation, for all GFP$^+$ cells with a clearly visible and identifiable leading process (44% of all GFP$^+$ cells at E17.5, 30% at P0.5, 26% at P1.5, 14% at P2.5). The 3D coordinates for each leading process were then exported into Excel, and their orientation was expressed in the brain's coordinate system (x=L−M, y=V−M, z=C−R), as a vector joining the start and end point of the process, using a custom Python script and the Pandas (*McKinney et al., 2010*) and Numpy (*van der Walt et al., 2011*) libraries. Each vector was defined by its orientation in spherical coordinates (polar and azimuthal angle) and overall length. Population level orientation data for the LGd, LP, and VP at E17.5 and P0 was plotted as heat-maps, by binning cells according to their spherical coordinates. The bins were then integrated along each axis to reveal a dominant orientation (e.g. for the LGd, 66% and 69% of cells oriented dorso-ventrally and caudo-rostrally, respectively). Polar histograms of leading process orientation in the dorsal-ventral-lateral-medial plane were also produced.

## Spatial clustering analysis

Unsupervised machine learning methods were used to investigate spatial organisation of *Sox14$^+$-Gad1$^+$* and *Sox14$^-$Gad1$^+$* cells. The 3D models of P14 *Sox14$^{GFP/+}$* thalamus generated with Neurolucida for NND analysis were again used to obtain the coordinates of all thalamic interneurons.

These data were analysed separately for each brain (n = 3) using a custom Python script, and partitioned into clusters using the k-means algorithm implemented in the library Scikit-Learn (*Buitinck et al., 2013*). The algorithm takes as input the expected number of clusters *k*.

Multiple values of *k* were tested, and evaluated using the silhouette coefficient metric of clustering performance (*Rousseeuw, 1987*), also implemented in Scikit-Learn. The silhouette coefficient is equal to the average ratio of distances between points within and between each cluster. More positive scores indicate coherent, well-separated clusters, whereas scores close to zero indicate overlapping clusters. The score was highest (0.472 ± 0.012) for k = 2, and the average fraction of all *Sox14$^+$* and *Sox14$^-$* cells in each of the resulting clusters was computed across all brains.

We also performed k-means clustering on the 3D distribution of *Gad1$^+$* cells obtained from ISH data from the Allen Mouse Brain Atlas. The silhouette score was again highest (0.512) for k = 2, and the resulting clusters have a spatial definition similar to those from the P14 *Sox14$^{GFP/+}$* thalamus.

## Statistics

### Comparison of distributions

The chi-squared test was used to test for significant differences in the thalamus-wide distribution of specific cell classes. This thalamus-wide comparison compensates for categorical errors arising from a degree of uncertainty in nuclear boundaries, as a result of variation in the sectioning plane and other factors.

For each distribution, average relative cell numbers were computed in Excel. A custom python script was used to compute the chi-squared statistic, and the corresponding p-value was computed using the chi-squared cumulative density function implemented in SciPy (*Jones et al., 2001*).

### Change in interneuron numbers in the Sox14 knockout

This was tested for statistical significance using unpaired two-sample two-tailed t-test, comparing the *Sox14* knockout to *Sox14$^{GFP/+}$* for each interneuron class separately (n = 3 brains/genotype).

Total interneuron numbers across all TC nuclei were compared and sampling was consistent between genotypes (each 10th thalamic section was analysed for each brain).

## Identification of outliers

In the analysis of the Marmoset's thalamus, the fraction of GAD1 single positive cells is low in most TC nuclei tested, this low frequency is partly due to the lower efficiency of the SOX14 probe compared to the GAD1 and therefore a systematic error. In the LD and MD, however, the frequency of GAD1 single positive cells is higher. To demonstrate that values for these two TC nuclei are outliers, we applied the method described in *Motulsky and Brown, 2006*, implemented in GraphPad Prism software.

## Acknowledgements

We thank the Wohl Cellular Imaging Centre, King's College London for support with imaging and image analysis software. We are grateful to Beatriz Rico, Monica Moissidis, and Patricia Hernandez at King's College London for the *Nkx2.1^{Cre}; Rosa26^{lsl-GFP}* and the *Lhx6^{Cre}; Rosa26^{lsl-GFP}* samples. We are grateful to the anonymous peer reviewers for their constructive and specific comments. This work was funded by the Biotechnology and Biological Sciences Research Council (BBSRC) grants BB/L020068/1 and BB/R007020/1 to AD and BB/R007659/1 to SB. Two-photon tomography was developed with Engineering and Physical Sciences Research Council (EPSRC) grant EP/J021199/1 to SRS.

## Additional information

### Funding

| Funder | Grant reference number | Author |
|---|---|---|
| Biotechnology and Biological Sciences Research Council | BB/L020068/1 | Alessio Delogu |
| Biotechnology and Biological Sciences Research Council | BB/R007020/1 | Alessio Delogu |
| Biotechnology and Biological Sciences Research Council | BB/R007659/1 | Stephen Brickley |
| Engineering and Physical Sciences Research Council | EP/J021199/1 | Simon R Schultz |
| Engineering and Physical Sciences Research Council | EP/L016737/1 | Gerald Moore |

The funders had no role in study design, data collection and interpretation, or the decision to submit the work for publication.

### Author contributions

Polona Jager, Conceptualization, Data curation, Software, Formal analysis, Investigation, Visualization, Methodology, Writing - original draft; Gerald Moore, Data curation, Software, Formal analysis, Methodology; Padraic Calpin, Data curation, Software, Methodology; Xhuljana Durmishi, Yoshiaki Kita, Yan Wang, Investigation, Methodology; Irene Salgarella, Data curation, Investigation; Lucy Menage, Dong Won Kim, Formal analysis, Investigation; Seth Blackshaw, Resources, Supervision; Simon R Schultz, Resources, Software, Supervision, Methodology; Stephen Brickley, Resources, Supervision, Funding acquisition; Tomomi Shimogori, Resources, Supervision, Methodology, Project administration; Alessio Delogu, Conceptualization, Data curation, Supervision, Funding acquisition, Investigation, Visualization, Methodology, Writing - original draft, Project administration, Writing - review and editing

### Author ORCIDs

Polona Jager ⓘ https://orcid.org/0000-0003-4484-9148
Irene Salgarella ⓘ https://orcid.org/0000-0003-3817-3321

Simon R Schultz http://orcid.org/0000-0002-6794-5813

Alessio Delogu https://orcid.org/0000-0002-4414-4714

### Ethics

Animal experimentation: Mice: Housing and experimental procedures were approved by the King's College London Ethical Committee and conformed to the regulations of the UK Home Office personal and project licences under the UK Animals (Scientific Procedures) 1986 Act. Marmoset: All experiments were conducted in accordance with the guidelines approved by the RIKEN Institutional Animal Care (W2020-2-022).

### Decision letter and Author response

Decision letter https://doi.org/10.7554/eLife.59272.sa1

Author response https://doi.org/10.7554/eLife.59272.sa2

## Additional files

### Supplementary files

• Transparent reporting form

### Data availability

All data generated or analysed during this study are included in the manuscript.

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
