## [Decision Letter]

**Acceptance summary:**

We thank you for submitting your work to *eLife* and apologize for the delay in the processing of your revised version. This work constitutes a very well presented and interesting study describing the ontogeny and origin of thalamic interneurones and relative contribution in mice and monkeys, with important relevance to brain evolution. In this revised version the authors have addressed most of the comments raised by the reviewers including a decisive description of the origin of *Sox14*-negative interneurons using an additional genetic model.

**Decision letter after peer review:**

Thank you for sending your article entitled "Dual midbrain and forebrain origins of thalamic inhibitory interneurons" for peer review at *eLife*. Your article has been evaluated by three peer reviewers, one of whom is a member of our Board of Reviewing Editors, and the evaluation has been overseen by Marianne Bronner as the Senior Editor.

Two of the three reviewers have outlined the quality of your article: "well presented and interesting study", "it carefully dissected the sources for mouse thalamic interneurons finding at least two major sources".

However, the appended reviews and off-line reviewer consultation raise several questions. In particular, all reviewers have highlighted the need of characterizing the origins of forebrain interneurons, especially by providing a more direct evidence of their prethalamic, hypothalamic or alternative origin. It would be essential to know if you can, by additional experiment, specifically address this point as well as potentially characterize the migratory streams associated.

Reviewer #1:

In this article by Jager et al., the authors investigate the developmental origin of thalamic interneurons, which play important physiological roles and display species-specific distributions in mammals. Previous work, including seminal work from this laboratory showed that thalamic interneurons in mice, which are largely restricted to visual first order thalamus, originate in the midbrain. Here the authors, explore in more depth the ontogeny of all thalamic interneurons in mice, as well as in new world monkeys, using a combination of fate-map tracing, automated quantitative reconstruction and histological analyses. They find that thalamic interneurons in mice have two distinct origins, one in the midbrain and a small population from the forebrain. This latter subpopulation was mostly detected as sparse interneurons in higher order thalamic nuclei. The authors further explored whether the two populations exist and changed in proportion/localization in new world monkeys, which display a widespread localization of interneurons (INs) in the thalamus. In both species, they found that Sox14+gad1+ INs constituted the largest population of INs, suggesting that the expansion of INs in primates is primarily due to an expansion in midbrain derived-INs. In addition, using intersectional genetics, they find that forebrain derived INs originate likely in the prethalamus and localize in the LD and MD.

Taken together, this is a very well presented and interesting study describing the ontogeny of thalamic INs and relative contribution in mice and monkeys, which benefits from an elegant automated approach to count INs and provides a compelling quantitative overview. In its present form, the study comprises some simplifications that should be addressed by the authors. The novelty of the findings and their significance is also not entirely clear, raising the question of whether this work should not be published in a more specialized journal that *eLife*.

1) One of the major findings of the article is the identification of a population of thalamic interneurons that does not derive from the midbrain. However, it's precise origin is determined indirectly, as originating from a *Dlx5+Nkx2.1*- source, likely the prethalamus. However, these cells could very well be produced by a *Nkx2.1*- region of the ganglionic eminences. It is relatively unclear why the authors support a prethalamic origin, apart from the presence of the marker parvalbumin (PV), which is not conserved across species. Since it was proposed in humans that some interneurons migrate from the ganglionic eminences, this specific point should be reinforced experimentally or better discussed.

2) A fundamental question that the article tackles is why are there much more interneurons in monkeys/primates than in rodents. While the authors discard that this would be due to the production of an entirely novel source of interneurons, they do not further address this important issue. One way to explore this would be to examine whether *Sox14+* interneurons form a single population of present subtypes? Do they originate from the same territory?

3) The orientation of migration of *Sox14+* midbrain-derived interneurons (presented in Figure 7 and Figure 7—figure supplement 1) constitutes an interesting novel result as it shows that interneurons present in the lateral nuclei might take a different route that the ones located in the VP. However, as presented, it is very difficult to understand what would be the routes taken by the interneurons. The authors could use the En1-cre (currently used for fate-map in Figure 6) at developmental stages to assess more precisely the migratory routes. It would also reinforce the fate-mapping since the label observed at P21 could be- in theory- linked to a postnatal late En1 expression in these cells.

4) Some of the counting are performed in adults (automated- Figure 1), P14 (Figures 2, 3, 5 and 8), P21 (Figure 6). While it might not be an issue if the system is stable from P14 onwards, the authors could clarify this point.

Reviewer #2:

The origins and developmental/genetic controls of thalamic interneurons are incompletely researched. I liked this paper because it carefully dissected the sources for mouse thalamic interneurons finding at least two major sources; En-Cre+Sox14+ midbrain and Dlx5/6-Cre+ Sox14- forebrain. I think that the paper could be published in *eLife*.

The only failing of the paper was that it did not establish whether the forebrain source is from the telencephalon or diencephalon/hypothalamus. This leaves unresolved the question of whether the mouse and human share the same forebrain origins of thalamic interneurons. In principle it should be easier in mouse, than in human, to firmly establish whether it is a telencephalic (e.g. ganglionic eminence) or diencephalic/hypothalamic source (e.g. prethalamus).

Did you carefully follow the potential migrations of the *Dlx5/6-^Cre+^* thalamic interneurons to get a hint about where they come from?

Below are some other specific suggestions/questions:

Introduction:

Intriguingly, we also find that in the mouse interneurons are organized in a spatial pattern according to their ontogeny, such that midbrain born interneurons are largely found in the principal sensory relays and modality related HO nuclei.

Do you mean FO nuclei? earlier you stated the midbrain origin FO interneurons.

Results:

Verified their previous fate mapping experiment: crossing En1-Cre with R26lsl-GFP reporter line and found interneurons labeled in the LGd, LP, VP, PO and MG.

Did you double label for *Sox14* and En-Cre;GFP?

Pv-Cre labeled MD and LD, but was absent from TC nuclei populated exclusively by *Sox14+* interneurons, such as the LGd and VP. P14 mouse 93.9% of *Pvalb+* cells in TC regions co-expressed GABA

Did you double label for *Sox14* and PV-Cre;GFP?

Regarding the references in the following sentence: “On the other hand, progenitor domains of the telencephalic medial ganglionic eminences (MGE) and preoptic area (POA) also generate *Pvalb+* interneurons, which are known to integrate in neocortical and hippocampal circuitries (Gelman et al., 2011; Lavdas et al., 1999; Wichterle et al., 2001; Xu et al., 2004; Xu et al., 2008),”

Please include the following reference for the requirement of Lhx6 expression in the MGE for the generation of PV+ interneurons. This could have import as there are Lhx6+ domains in the hypothalamus that in principle could be a source of PV+ thalamic interneurons

Distinct molecular pathways for development of telencephalic interneuron subtypes revealed through analysis of Lhx6 mutants. Zhao et al., 2008.

Regarding the *Dlx5-^Cre^* (Monory et al., 2006): this allele should be referred to as *Dlx5/6-^Cre^*the Cre gene was placed under the control of the I56i and I56ii intergenic enhancer sequences (Zerucha et al., 2000).

Note, my lab has had trouble with this *Dlx5/6-^Cre^* allele because it has ectopic activity from time to time. I recommend repeating the experiment with another *Dlx-^Cre^* mouse, such as the *DlxI12b-^Cre^* (Generation of Cre-transgenic mice using *Dlx1/Dlx2* enhancers and their characterization in GABAergic interneurons. Potter GB, Petryniak MA, Shevchenko E, McKinsey GL, Ekker M, Rubenstein JL. Mol Cell Neurosci. 2009 Feb;40(2):167-86.)

The authors could not determine whether the source of the forebrain-derived thalamic interneurons was from the telencephalon (perhaps the CGE), or from the prethalamus. There are some Cre lines that may be helpful in further dissection of this problem – see lines 599 and 1060 in: Subpallial Enhancer Transgenic Lines: a Data and Tool Resource to Study Transcriptional Regulation of GABAergic Cell Fate. Silberberg SN, Taher L, Lindtner S, Sandberg M, Nord AS, Vogt D, Mckinsey GL, Hoch R, Pattabiraman K, Zhang D, Ferran JL, Rajkovic A, Golonzhka O, Kim C, Zeng H, Puelles L, Visel A, Rubenstein JLR. Neuron. 2016 Oct 5;92(1):59-74.

Reviewer #3:

This is a follow-up study after Jager et al (2016), which was focused on Sox14+ GABAergic neurons in the LGv. In this study, the authors investigate origins of both *Sox14+* and *Sox14-* GABAergic neurons settled in a larger thalamic region by mouse genetic lineage analyses. I am not particularly enthusiastic about publishing this paper in *eLife*, but at the same time, do not strongly oppose to a decision of acceptance after some revisions. Unfortunately, the manuscript contains few novel influential data, but covers important topics and takes a standard approach in developmental neurobiology.

1) This study is a mosaic collection of small experiments which are related to each other only weakly. The manuscript contains state-of-the-art imaging and quantification techniques, marmoset histological data, and lineage data using multiple different mouse lines. The authors attempt to put these results in a story of evolutionary increase of GABAergic interneurons in the thalamus. However, their attempt seems not successful. Simply based on their data, the increase of GABAergic interneurons through mammalian evolution is achieved by expansion of midbrain-derived *Sox14+* GABAergic neurons. However, in this paper, the authors do not clearly claim so, but instead, seem to support a previous hypothesis that the thalamic evolution in primates is achieved by a long-distance influx of GABAergic neurons from the ganglionic eminence. This twisted logic is very confusing and should be corrected. In the Abstract, they claim that "thalamic interneurons can be studied in mice" but this claim cannot be accepted as the conclusion of this study.

2) The forebrain origin of *Sox14-* GABAergic neurons is not soundly supported in this study. The use of *Pvalb* as a marker of *Sox14-* GABAergic neurons is not convincing. The authors claim that the majority of *Pvalb+* neurons are GABAergic, but do not show the proportion of GABAergic neurons that express *Pvalb*. The authors should provide more convincing justification that *Pvalb* specifically marks *Sox14*- GABAergic neurons in the thalamus proper (excluding RT neurons, because they are just ignored in other analyses.) Do Sox14+ GABAergic neurons express *Pvalb*? I wonder why the authors do not simply examine the expression of GAD in *Dlx5* or *Nkx2.1* lineage tracing?

3) Only in the end of this study, the authors reveal that "the forebrain origin" indicates the prethalamic progenitors but not the ganglionic eminence. This is quite misleading and also lacks actual experimental support. First and foremost, the prethalamic origin of GABAergic neurons in the thalamus is simply natural and has been shown experimentally for years before (e.g. Inamura et al., 2011).

---

## [Author Response]

Reviewer #1:[…] In its present form, the study comprises some simplifications that should be addressed by the authors. The novelty of the findings and their significance is also not entirely clear, raising the question of whether this work should not be published in a more specialized journal that eLife.1) One of the major findings of the article is the identification of a population of thalamic interneurons that does not derive from the midbrain. However, it's precise origin is determined indirectly, as originating from a Dlx5+Nkx2.1- source, likely the prethalamus. However, these cells could very well be produced by a Nkx2.1- region of the ganglionic eminences. It is relatively unclear why the authors support a prethalamic origin, apart from the presence of the marker parvalbumin (PV), which is not conserved across species. Since it was proposed in humans that some interneurons migrate from the ganglionic eminences, this specific point should be reinforced experimentally or better discussed.

We now provide direct evidence for a prethalamic origin of the newly described population of thalamic interneurons. To achieve this, we have performed fate mapping of prethalamic and hypothalamic progenitors with the *Foxd1-Cre* mouse line. Developmental expression and function of *Foxd1* and applications of *Foxd1-Cre* had been previously published by the Blackshaw lab – importantly, this line is not expressed in the ganglionic eminences of the telencephalon. We have therefore tested whether the forebrain-derived thalamic interneuron class that we have identified derives from *Foxd1^+^* progenitors. This is indeed the case, with at least 85% of parvalbumin cells in thalamocortical nuclei labelled by the fluorescent reporter (Figure 8E,F,K,L). Weak reporter expression and potential incomplete recombination are a likely explanation for the residual 15% of thalamic parvalbumin cells that were not labelled with the reporter in our experiments.

We have then refined further the source of this thalamic interneuron class by complementing the previously presented fate mapping with *Nkx2.1-Cre* with new fate mapping using the *Lhx6-Cre* line. *Nkx2.1* and *Lhx6* are broadly associated with GABAergic neurogenesis in the hypothalamus and we now show that none of these genetically defined GABA progenitor domains contribute to the parvalbumin thalamic interneuron class. Also, the characterisation of the fate mapping with the *Lhx6-Cre* further confirms that the telencephalic ganglionic eminences are an unlikely source for this thalamic interneuron class (Figure 8G,H,I,J,K).

2) A fundamental question that the article tackles is why are there much more interneurons in monkeys/primates than in rodents. While the authors discard that this would be due to the production of an entirely novel source of interneurons, they do not further address this important issue. One way to explore this would be to examine whether Sox14+ interneurons form a single population of present subtypes? Do they originate from the same territory?

The evidence we have at hand from previous exploratory work with focal electroporation suggests that the *Sox14^+^* interneurons that populate TC nuclei (e.g. the visual and VB complex) arise all in a rostral domain of the dorsal midbrain territory that falls within the prospective superior colliculus (SC) in standard anatomy atlases (e.g. Allen Brain reference atlas). Targeting of the more caudal portion of the midbrain region still labelled the midbrain but did not result in any thalamic interneuron labelling. Hence, the precise source territory of the *Sox14^+^* thalamic interneurons may be the tectal gray, however, this subdivision is not yet fully integrated in standard anatomy atlases. Additionally, all *Sox14^+^* interneurons expressed markers that had been described in LGd interneurons previously (Otx2 and Chrna6; see also Golding et al., 2014 and our manuscript’s Figure 3—figure supplement 1). Of note, a recent preprint by Bakken et al., 2020 that used scRNAseq from mouse, macaque and humans is consistent with our ontogenetic model and extends its validity to macaques and humans (https://www.biorxiv.org/content/10.1101/2020.11.05.367482v2).

3) The orientation of migration of Sox14+ midbrain-derived interneurons (presented in Figure 7 and Figure 7—figure supplement 1) constitutes an interesting novel result as it shows that interneurons present in the lateral nuclei might take a different route that the ones located in the VP. However, as presented, it is very difficult to understand what would be the routes taken by the interneurons. The authors could use the En1-cre (currently used for fate-map in Figure 6) at developmental stages to assess more precisely the migratory routes. It would also reinforce the fate-mapping since the label observed at P21 could be- in theory- linked to a postnatal late En1 expression in these cells.

Indeed, the analysis we presented stems from the assessment of the migratory morphology and location at developmental stages (E16.5, E17.5, P0.5, P1.5 and P2.5), with the *En1-Cre* line. Quantitative analysis is only technically possible when labelled cells leave the strongly labelled midbrain territory and enter pretectal territory *en route* to the thalamus. These developmental time points are summarised in the manuscript and in Figure 7 and figure 7—figure supplement 1. Future work with clonal labelling may provide further detail of the migration within the *En1-Cre^+^* midbrain territory.

4) Some of the counting are performed in adults (automated- Figure 1), P14 (Figures 2, 3, 5 and 8), P21 (Figure 6). While it might not be an issue if the system is stable from P14 onwards, the authors could clarify this point.

We have improved the text to make clear that also the automated quantification presented in Figure 1 is done at P21. We have not observed any evidence that would suggest the system undergoes significant changes between P14 and P21.

Reviewer #2:[…] The only failing of the paper was that it did not establish whether the forebrain source is from the telencephalon or diencephalon/hypothalamus. This leaves unresolved the question of whether the mouse and human share the same forebrain origins of thalamic interneurons. In principle it should be easier in mouse, than in human, to firmly establish whether it is a telencephalic (e.g. ganglionic eminence) or diencephalic/hypothalamic source (e.g. prethalamus).Did you carefully follow the potential migrations of the Dlx5/6-Cre+ thalamic interneurons to get a hint about where they come from?

We have addressed the important question of where the *Dlx5/6-Cre^+^* thalamic interneurons are coming from by using the prethalamic/hypothalamic *Foxd1-Cre* line and then further spatially refined the *Dlx5/6^+^Foxd1^+^* domain using the *Lhx6-Cre* to complement the previously used *Nkx2.1-Cre* line. As detailed in our reply to point 1 of reviewer 1, we can now more confidently say that the *Dlx5/6-Cre*^+^ thalamic interneurons come from the prethalamus (or in genetic terms, the *Dlx5^+^Foxd1^+^Nkx2.1^-^Lhx6^-^* rostral diencephalon). Importantly, the use of the *Lhx6-Cre* line, adds further evidence against a ganglionic origin for thalamic interneurons in the mouse. Figure 8 has been redesigned to present these novel data and the text modified accordingly.

Below are some other specific suggestions/questions:Introduction:Intriguingly, we also find that in the mouse interneurons are organized in a spatial pattern according to their ontogeny, such that midbrain born interneurons are largely found in the principal sensory relays and modality related HO nuclei.Do you mean FO nuclei? – earlier you stated the midbrain origin FO interneurons.

We intended to refer to the presence of midbrain-born interneurons in the first and higher order sensory parts of the thalamus (visual, somatosensory, mostly, but also auditory) but not higher order limbic nuclei. For improved clarity the text is now corrected to refer to “sensory relays”, only.

Results:Verified their previous fate mapping experiment: crossing En1-Cre with R26lsl-GFP reporter line and found interneurons labeled in the LGd, LP, VP, PO and MG.Did you double label for Sox14 and En-Cre;GFP?

There are no commercial antibodies for Sox14 that work in our hands. Under circumstances where GFP expression is very high, we have successfully combined IHC for GFP with ISH for a second marker’s mRNA and quantified double positive neurons, however the GFP expression level from the R26 locus is too weak to give good results using this approach. An alternative approach would be to generate a triple transgenic *Sox14GFP,En1-Cre,R26lsl-Tomato*, but this required lengthy breeding for the timeline of this revision. However, it should be stressed that the dorsal midbrain territory generating the *En1-Cre^+^* interneurons is strongly *Sox14^+^* and that the pattern of labelling with *Sox14GFP* or *En1-Cre* fate mapping in thalamocortical nuclei is indistinguishable. Given the sparsity of interneurons in the mouse thalamus overall, we are confident that the *En1-Cre^+^* and the *Sox14^+^* interneurons are the same cells. This is supported also by our previous data, where we have shown that blocking GABAergic neurogenesis in the midbrain using an *En1-Cre*,*Gata2^flox/flox^* strategy, eliminated the *Sox14* interneurons in the thalamus (Jager et al., 2016).

Pv-Cre labeled MD and LD, but was absent from TC nuclei populated exclusively by Sox14+ interneurons, such as the LGd and VP. P14 mouse 93.9% of Pvalb+ cells in TC regions co-expressed GABADid you double label for Sox14 and PV-Cre;GFP?

We have addressed this question by double labelling Pvalb/GFP in the *En1-Cre,R26lsl-GFP* line or the *Sox14GFP* line. In both lines there were 0 double positive cells out of hundreds of Pvalb^+^ counted, confirming that the midbrain-derived interneurons do not express Pvalb. Given the complete absence of double positive cells, we did not see the benefit of preparing a new figure, but have entered the results of the counts for both reporter lines.

Regarding the references in the following sentence: “On the other hand, progenitor domains of the telencephalic medial ganglionic eminences (MGE) and preoptic area (POA) also generate Pvalb+ interneurons, which are known to integrate in neocortical and hippocampal circuitries (Gelman et al., 2011; Lavdas et al., 1999; Wichterle et al., 2001; Xu et al., 2004; Xu et al., 2008),”Please include the following reference for the requirement of Lhx6 expression in the MGE for the generation of PV+ interneurons. This could have import as there are Lhx6+ domains in the hypothalamus that in principle could be a source of PV+ thalamic interneuronsDistinct molecular pathways for development of telencephalic interneuron subtypes revealed through analysis of Lhx6 mutants. Zhao et al., 2008.

We have followed up on this suggestion and performed new fate mapping with the *Lhx6-Cre* line. There are no *Lhx6-Cre^+^* cells in the thalamus, confirming that the MGE as well as other hypothalamic GABAergic domains do not contribute interneurons to the thalamus (Figure 8I,J,K). We have added the reference from Zhao et al.

Regarding the Dlx5-Cre (Monory et al., 2006): this allele should be referred to as Dlx5/6-Cre the Cre gene was placed under the control of the I56i and I56ii intergenic enhancer sequences (Zerucha et al., 2000).

This has now been corrected throughout the manuscript.

Note, my lab has had trouble with this Dlx5/6-Cre allele because it has ectopic activity from time to time. I recommend repeating the experiment with another Dlx-Cre mouse, such as the DlxI12b-Cre (Generation of Cre-transgenic mice using Dlx1/Dlx2 enhancers and their characterization in GABAergic interneurons. Potter GB, Petryniak MA, Shevchenko E, McKinsey GL, Ekker M, Rubenstein JL. Mol Cell Neurosci. 2009 Feb;40(2):167-86.)

We have also noticed occasional ectopic Cre activity and discussed it in the main text (page 11, line 22 onward) and in Figure 8—figure supplement 1. In our samples, this appears as rare events that lead to small clones of spatially clustered, mostly glia, cells. We didn’t see this as affecting the validity of our findings, as Pvalb^+^/GABA^+^ thalamic interneurons are consistently labelled by the *Dlx5/6-Cre* line, while ectopically labelled cells are rare, vary in nature, number and anatomical location. Having now performed new *Foxd1-Cre* fate mapping, we add independent validation of the *Dlx5/6-Cre* result.

The authors could not determine whether the source of the forebrain-derived thalamic interneurons was from the telencephalon (perhaps the CGE), or from the prethalamus. There are some Cre lines that may be helpful in further dissection of this problem – see lines 599 and 1060 in: Subpallial Enhancer Transgenic Lines: a Data and Tool Resource to Study Transcriptional Regulation of GABAergic Cell Fate. Silberberg SN, Taher L, Lindtner S, Sandberg M, Nord AS, Vogt D, Mckinsey GL, Hoch R, Pattabiraman K, Zhang D, Ferran JL, Rajkovic A, Golonzhka O, Kim C, Zeng H, Puelles L, Visel A, Rubenstein JLR. Neuron. 2016 Oct 5;92(1):59-74.

We are grateful for this suggestion and we have added fate mapping with two additional *Cre* lines, one restricted to the diencephalic prethalamus/hypothalamus (*Foxd1-Cre*) and one restricted to MGE and part of the GABAergic hypothalamus (*Lhx6-Cre*).

Reviewer #3:This is a follow-up study after Jager et al (2016), which was focused on Sox14+ GABAergic neurons in the LGv. In this study, the authors investigate origins of both Sox14+ and Sox14- GABAergic neurons settled in a larger thalamic region by mouse genetic lineage analyses. I am not particularly enthusiastic about publishing this paper in eLife, but at the same time, do not strongly oppose to a decision of acceptance after some revisions. Unfortunately, the manuscript contains few novel influential data, but covers important topics and takes a standard approach in developmental neurobiology.1) This study is a mosaic collection of small experiments which are related to each other only weakly. The manuscript contains state-of-the-art imaging and quantification techniques, marmoset histological data, and lineage data using multiple different mouse lines. The authors attempt to put these results in a story of evolutionary increase of GABAergic interneurons in the thalamus. However, their attempt seems not successful. Simply based on their data, the increase of GABAergic interneurons through mammalian evolution is achieved by expansion of midbrain-derived Sox14+ GABAergic neurons. However, in this paper, the authors do not clearly claim so, but instead, seem to support a previous hypothesis that the thalamic evolution in primates is achieved by a long-distance influx of GABAergic neurons from the ganglionic eminence. This twisted logic is very confusing and should be corrected. In the Abstract, they claim that "thalamic interneurons can be studied in mice" but this claim cannot be accepted as the conclusion of this study.

In this paper we present the first comprehensive description of the ontogeny of thalamic interneurons in the mouse. Given the surprise finding that about 80% of thalamic interneurons are specified outside of the forebrain, we have aimed to verify whether such peculiar ontogeny is unique of the rodent or shared also with species that are known to have some of the highest proportions of thalamic interneurons, among mammals. Based on the marmoset study, we propose that the midbrain programme for thalamic interneurons is likely the dominant throughout mammals.

The discovery of an additional class of forebrain-derived thalamic interneurons in the mouse, required that we addressed the possibility of its homology to the thalamic interneuron lineage described by Rakic and colleagues, in humans, as MGE-derived. We have taken multiple approaches (*Dlx5/6-Cre, Foxd1-Cre, Nkx2.1-Cre and Lhx6-Cre*) to demonstrate that this mouse interneuron class is not homologous to thehuman one.

While, humans appear to have evolved an additional interneuron class, we propose a model whereby the dual forebrain and midbrain origin of thalamic interneurons is the shared plan for mammals, regardless of the density of interneurons in their thalamocortical regions. This conceptual framework and the validity of the concluding remark of our Abstract are reinforced by a recent preprint by Bakken et al., 2020 at the Allen Brain Institute (https://www.biorxiv.org/content/10.1101/2020.11.05.367482v2 see Supplementary Figure 5B-D), which confirms the presence of two major classes of thalamic interneurons, one labelled by *Sox14* and the other by *Dlx* genes, in mouse and primates.

2) The forebrain origin of Sox14- GABAergic neurons is not soundly supported in this study. The use of Pvalb as a marker of Sox14- GABAergic neurons is not convincing. The authors claim that the majority of Pvalb+ neurons are GABAergic, but do not show the proportion of GABAergic neurons that express Pvalb. The authors should provide more convincing justification that Pvalb specifically marks Sox14- GABAergic neurons in the thalamus proper (excluding RT neurons, because they are just ignored in other analyses.) Do Sox14+ GABAergic neurons express Pvalb? I wonder why the authors do not simply examine the expression of GAD in Dlx5 or Nkx2.1 lineage tracing?

We should clarify that we do not use Pvalb expression to prove forebrain origin. The forebrain-origin is based on the use of the *Dlx5/6-Cre* and *Foxd1-Cre* lines. Virtually all *Dlx5/6-Cre^+^* and *Foxd1-Cre*^+^ GABA neurons in thalamocortical nuclei also express Pvalb, so Pvalb is a convenient additional marker. For instance, we plan in the future to use *Pvalb-Cre* to study functional properties of these interneurons and other labs may want to use this information too. All quantifications reported in the manuscript exclude the RT and only focus on cells contained within thalamocortical nuclei. We have added the results showing complete lack of co-expression between Pvalb and Sox14 in the *Sox14GFP* and *En1-Cre* lines. We did examine expression of GAD (GABA IHC) in the *Dlx5/6-Cre* and in the *Foxd1-Cre* lines (see Figure 8B, F, L and related text).

3) Only in the end of this study, the authors reveal that "the forebrain origin" indicates the prethalamic progenitors but not the ganglionic eminence. This is quite misleading and also lacks actual experimental support. First and foremost, the prethalamic origin of GABAergic neurons in the thalamus is simply natural and has been shown experimentally for years before (e.g. Inamura et al., 2011).

Indeed, the prethalamus is a well-known source of GABAergic neurons and Inamura et al., 2011 nicely highlights the prethalamic contribution to RT, ZI and part of the LGv. The *Olig2* and *Olig3* lines used in that study do not have sufficient spatial specificity to address precisely the origin of thalamic interneurons as they are expressed in the telencephalon and diencephalon, including the second prosomere (see, for instance, Vue et al., J. Comp. Neuro. 2007; 505:73–91 – Figures 8 and 9). The novelty of our finding is that the prethalamus, surprisingly and despite being the obvious candidate, does not generate the vast majority of thalamic interneurons, which are instead specified in the midbrain. The finding from Rakic and colleagues, specifically for the human thalamic interneurons is also surprising, as it shows that thalamic interneurons can come from more distant GABAergic neurogenic territories than the prethalamus. The experimental support for the prethalamic origin in the mouse in our study is now reinforced by the addition of *Foxd1-Cre* fate mapping, complementing the *Dlx5/6-Cre* line.